# High carbon dioxide emissions from Australian estuaries driven by geomorphology and climate

Jacob Z.-Q. Yeo [1] ✉, Judith A. Rosentreter [1], Joanne M. Oakes [1], Kai G. Schulz [1] & Bradley D. Eyre [1]

Estuaries play an important role in connecting the global carbon cycle across the land-to-ocean continuum, but little is known about Australia's contribution to global $CO_2$ emissions. Here we present an Australia-wide assessment, based on $CO_2$ concentrations for 47 estuaries upscaled to 971 assessed Australian estuaries. We estimate total mean (±SE) estuary $CO_2$ emissions of $8.67 \pm 0.54$ Tg $CO_2$-C yr$^{-1}$, with tidal systems, lagoons, and small deltas contributing 94.4%, 3.1%, and 2.5%, respectively. Although higher disturbance increased water-air $CO_2$ fluxes, its effect on total Australian estuarine $CO_2$ emissions was small due to the large surface areas of low and moderately disturbed tidal systems. Mean water-air $CO_2$ fluxes from Australian small deltas and tidal systems were higher than from global estuaries because of the dominance of macrotidal sub-tropical and tropical systems in Australia, which have higher emissions due to lateral inputs. We suggest that global estuarine $CO_2$ emissions should be upscaled based on geomorphology, but should also consider land-use disturbance, and climate.

Estuaries play an important role connecting the carbon cycle across the land-to-ocean aquatic continuum, processing large amounts of allochthonous and autochthonous carbon[1]. This is despite estuaries constituting only a small fraction of the world's surface (0.2%)[2] compared to continental shelf seas (5%)[3] and the open ocean (64%)[4]. Carbon from upstream rivers and associated coastal wetlands entering estuaries is either buried (and potentially stored long-term), emitted to the atmosphere in the form of greenhouse gases, or exported to the ocean[5,6]. Estuarine $CO_2$ emissions are estimated to equate to the size of the $CO_2$ sink in shelf seas ($0.268 \pm 0.225$ Pg C yr$^{-1}$), or 19% of the $CO_2$ sequestration in the open ocean[7], but early estimates were mostly based on studies in the northern hemisphere (e.g. refs. 5,6,8,9). $CO_2$ emissions from estuaries can differ between estuary geomorphic types[10,11], but the mechanisms by which geomorphology affects estuarine $CO_2$ emissions in Australia have not been determined. There is also limited knowledge of how disturbance impacts $CO_2$ emissions and how different geomorphic estuary types modify any disturbance effect.

Low to high disturbance and land-use changes in the upper catchment have the potential to alter the quantity and quality of carbon delivered to estuaries[5,12], and hence the associated estuary $CO_2$ emissions[12,13]. Dissolved inorganic carbon (DIC)[14], dissolved organic carbon (DOC)[15], and particulate organic carbon (POC)[16] inputs typically increase in impacted estuaries and tend to be associated with increased $CO_2$ emissions[7,17,18]. However, the effect of land-use on $CO_2$ emissions from estuaries can vary[12,19]. For instance, moderately and highly disturbed estuaries in Australia have been reported to emit more $CO_2$ per unit area ($37 \pm 10$ mmol $CO_2$-C m$^{-2}$ d$^{-1}$) than less disturbed estuaries ($6.3 \pm 4$ mmol $CO_2$-C m$^{-2}$ d$^{-1}$)[12], whereas very small coastal estuaries with high land-use changes (>90% of catchment modified) had lower $CO_2$ emissions than estuaries with low land-use changes (~21% of catchment modified)[15]. Moreover, changes in land-use along an estuarine gradient can influence nutrient cycling (e.g., decomposition) and change the quantity and quality (labile or refractory) of organic matter inputs[20–22], resulting in increases or decreases

[1]Centre for Coastal Biogeochemistry, Faculty of Science and Engineering, Southern Cross University, PO Box 157, East Lismore, NSW 2480, Australia. ✉e-mail: jacob.yeo@scu.edu.au

in $CO_2$ emissions between riverine-upstream, mid-estuary, and near-marine regions[23,24].

Estuaries of different geomorphology are the result of varying influences of river discharge, tidal amplitude, and wave energy, which determine estuarine hydrological characteristics such as water depth, current velocity, and water residence times[2,25,26]. In turn, water depth and current velocity influence the gas transfer velocity ($k$) which controls the rate of $CO_2$ emission from the water into the atmosphere[27–29]. Water residence times control $CO_2$ emissions in estuaries by determining the direction and intensity of estuarine water-air $CO_2$ fluxes[2,7], because long water residence times allow for more carbon decomposition, resulting in increased DIC that can be emitted as $CO_2$[7,28,30]. Shorter water residence times accelerate DOC export to the ocean, resulting in lower $CO_2$ emissions[17,18]. Stratification of the water column can also influence $CO_2$ emissions[31,32]. Photosynthetic $CO_2$ uptake occurs in the surface layer, whereas $CO_2$ respired in the bottom waters is isolated from atmospheric exchange at the surface[32–34], resulting in an overall increase in $CO_2$ partial pressure ($p$CO$_2$) but unaffected water-air $CO_2$ flux rates[32,34]. Stratification occurs in estuaries with weak tidal forcing, which leads to the separation of water layers of different densities (i.e. salinities)[31,35] or temperatures (thermohaline stratification)[32–34]. In estuaries with stronger tidal influence, tidal pumping can also increase $CO_2$ emissions through the lateral import of DIC and DOC from coastal wetlands to the estuary[36,37]. Tidal pumping can be a significant driver of $CO_2$ emissions, with groundwater-derived export accounting for 93% to 99% of DIC and 89% to 92% of DOC exported from mangroves into tidal creeks[36]. In very large river systems[2] such as the Amazon River[38] in Brazil and the

Yangtze[39] River in China, riverine transport from the estuary to the ocean can result in extensive estuarine plumes that can act as either a source or a sink of $CO_2$. However, such systems do not exist in Australia. A recent global analysis showed that fjords predominantly act as $CO_2$ sinks, while tidal systems and deltas emit more $CO_2$ than lagoons[11].

Australia makes a significant contribution to the total number and surface area of estuaries globally. Australia's coastline of 36,700 km includes 971 estuaries[40], accounting for 1.82% of the total number of estuaries globally[41] and 2.35% of global estuarine surface area[2]. More importantly, the majority of Australia's estuaries (70.6%) are classified as low or moderately disturbed[40], contrasting with predominantly disturbed estuaries in Europe and the United States where the majority of estuarine $CO_2$ emissions have been measured[9,10]. This reflects Australia's population density of only 3.3 persons km$^{-2}$[42], the 3rd lowest in the world. Despite Australia's contribution to global estuary number and surface area, $CO_2$ emissions have been measured in only a few Australian estuaries (e.g. refs. 12,23,43,), and there are no estimates of total $CO_2$ emissions from Australian estuaries. In this study, we (1) calculated $CO_2$ emissions from $p$CO$_2$ measurements from 36 estuaries in different climate zones in Australia and combined these $CO_2$ emission estimates with previously published $CO_2$ emissions from 11 other Australian estuaries[12,30] (total of 47 estuaries), and (2) assessed the interaction effects of anthropogenic disturbance and geomorphology on $CO_2$ emissions for these 47 estuaries. Based on disturbance and geomorphology classifications of Australian estuaries, we then scaled up $CO_2$ emissions from the 47 estuaries to all 971 assessed estuaries to better constrain Australia's contribution to global estuarine $CO_2$ emissions. We hypothesised that estuarine geomorphic type and

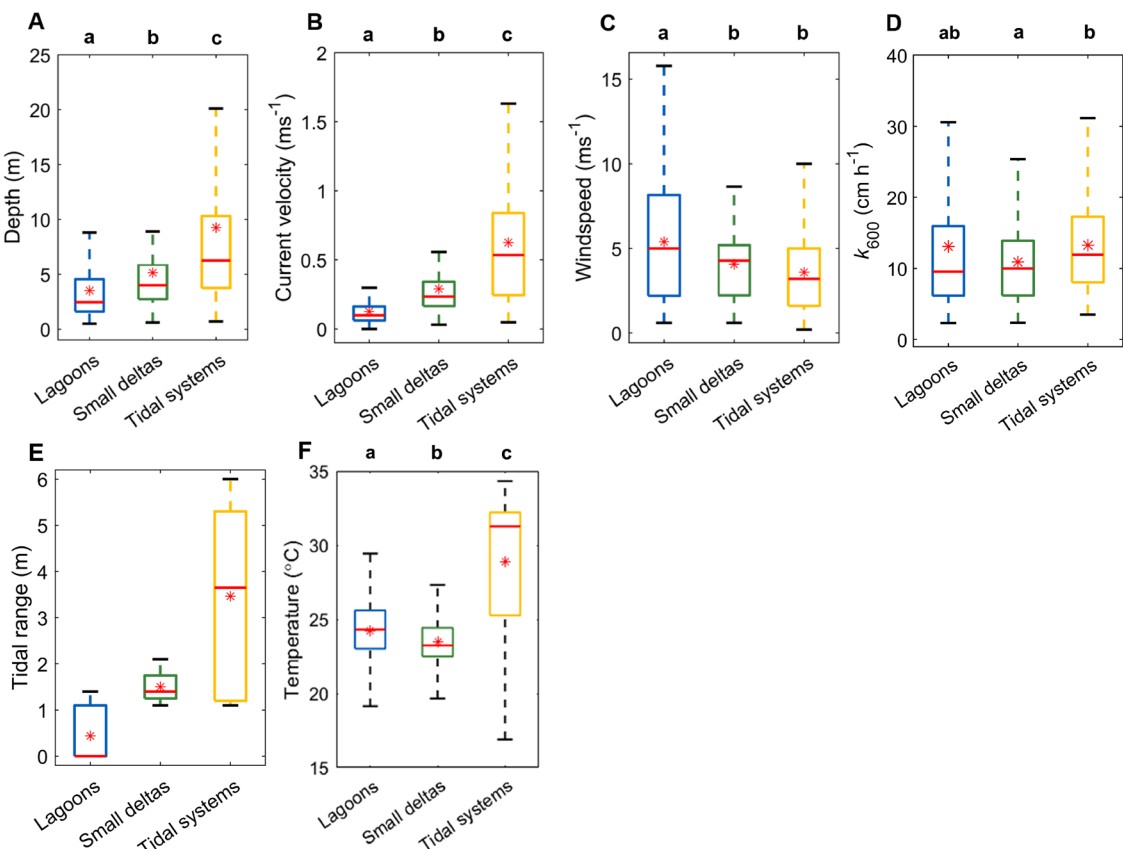

**Fig. 1 | Physical parameters in the three estuary types.** Median (red line), mean (red asterisk), 1st and 3rd interquartile ranges (box caps), minimum, and maximum values (whiskers) of (**A**) water depth, (**B**) current velocity, (**C**) wind speed, (**D**) mean gas transfer velocity normalised to Schmidt no. 600 ($k_{600}$) calculated from the five parameterisations (Table 2), (**E**) tidal range, and (**F**) temperature in the lagoons (blue, $n$: **A** = 92, **B** = 101, **C** = 88, **D** and **F** = 3789, and **E** = 21), small deltas (green, $n$:

**A** = 75, **B** = 67, **C** = 79, **D** = 3362, **E** = 12, and **F** = 3622), and tidal systems (yellow, $n$: **A** = 121, **B** = 112, **C** = 126, **D** = 5207, **E** = 14, and **F** = 5720). Outliers were omitted from the graphs. Letters above figures denote statistical differences among estuary types, with letters that are the same indicating no significant difference (PERMANOVA, two-tailed, and at 95% confidence interval). Source data are provided as a Source Data file.

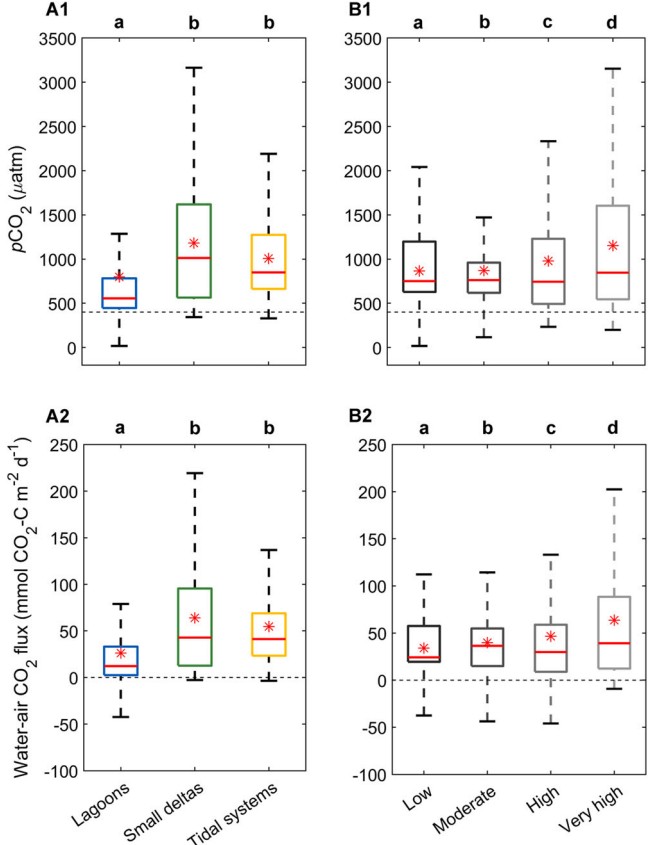

**Fig. 2 | CO₂ partial pressure ($p$CO₂) and water-air CO₂ flux in estuary types and disturbance groups.** Median (red line), mean (red asterisk), 1st and 3rd interquartile ranges (box caps), minimum and maximum values (whiskers) of $p$CO₂ and water-air CO₂ flux at per-minute resolution in the (**A**) lagoons (blue, $n = 3789$), small deltas (green, $n = 3622$), and tidal systems (yellow, $n = 5720$); and (**B**) low (black, $n = 1796$), moderate (dark grey, $n = 3189$), high (grey, $n = 3677$), and very high (light grey, $n = 4469$) disturbance groups. Outliers were omitted from the figures. Dotted line along the x-axis represents atmospheric $p$CO₂ and water-air flux CO₂ equilibrium. Letters above figures denote statistical differences among estuary types, with letters that are the same indicating no significant difference (PERMANOVA, two-tailed, and at 95% confidence interval). Source data are provided in the Source Data file.

disturbance level (including land-use change) would significantly impact estuarine water column $p$CO₂ and CO₂ emissions, and that there would be an interaction between geomorphic type, disturbance level, and CO₂ emissions. We further hypothesised that relative CO₂ emissions from Australian estuaries would be lower than global estuary CO₂ emissions because of generally lower disturbance found in estuaries in Australia.

## Results

### Physical differences between estuary types

Mean (min-max) tidal range was highest in tidal systems ($n = 14$) (3.7 m (1.1–6.0 m)), moderate in small deltas ($n = 12$) (1.5 m (1.1–2.1 m)), and generally lower in lagoons ($n = 21$) (0.6 m (0–1.4 m)) (Fig. 1E). Water depth ($n = 92$, 75, and 121, respectively) (Fig. 1A) and current velocity ($n = 101$, 67, and 112, respectively) (Fig. 1B) significantly increased ($p = 0.001$) from lagoons to small deltas to tidal systems. Wind speed was significantly higher in lagoons ($n = 88$) than in small deltas ($n = 79$) ($p = 0.004$) and tidal systems ($n = 126$) ($p = 0.001$), but was similar between small deltas and tidal systems ($p = 0.137$) (Fig. 1C). The mean gas transfer velocity normalised to the Schmidt number of 600 ($k_{600}$), was highest in tidal systems and significantly lower in small deltas

($p = 0.001$). Although lagoons had the lowest $k_{600}$ ($n = 751$), it was not significantly different from small deltas ($n = 667$) and tidal systems ($n = 1036$) (negative t-values) (Fig. 1D and Supplementary Fig. 1). Tidal range significantly increased from the lowest in lagoons ($n = 21$) to the highest in tidal systems ($n = 14$, small deltas: $n = 12$) ($p = 0.001$) (Fig. 1E). Temperature differed significantly between estuary types (lagoons: $n = 751$) ($p = 0.001$) with the lowest mean temperature in small deltas ($n = 719$) and the highest mean temperature in tidal systems ($n = 1138$) (Fig. 1F and Supplementary Table 3). Across all estuary types, except tidal systems, and within estuary types, temperature did not significantly correlate with $p$CO₂ and water-air CO₂ flux. In tidal systems, there was a significant increase in water-air CO₂ flux with temperature ($r = 0.254$, $p = 0.01$).

### Estuary $p$CO₂ and water-air CO₂ fluxes

The majority of the estuaries studied were a source of CO₂ to the atmosphere (Fig. 2A). Mean (±SE) $p$CO₂ and water-air CO₂ fluxes were $799 \pm 13$ μatm and $26.4 \pm 0.9$ mmol CO₂-C m⁻² d⁻¹ in lagoons ($n = 751$), $1181 \pm 12$ μatm and $63.9 \pm 1.1$ mmol CO₂-C m⁻² d⁻¹ in small deltas ($n = 719$), and $1007 \pm 6$ μatm and $54.8 \pm 0.8$ mmol CO₂-C m⁻² d⁻¹ in tidal systems ($n = 1138$) (Table 1). Six small deltas and three tidal systems had small sections that were weak CO₂ sinks (>−3.5 mmol CO₂-C m⁻² d⁻¹). Four lagoons were overall CO₂ sinks, whereas 14 lagoons had sections that were strong CO₂ sinks (up to −64.7 mmol CO₂-C m⁻² d⁻¹). Although $p$CO₂ and water-air CO₂ fluxes in the lagoons had the largest range, $p$CO₂ and water-air CO₂ fluxes were significantly lower than in small deltas and tidal systems ($p = 0.001$), with lower means and medians (Fig. 2A). $p$CO₂ ($p = 0.693$) and water-air CO₂ fluxes ($p = 0.064$) in small deltas were not significantly different to those in tidal systems (Fig. 2A).

### Disturbance effects on estuary CO₂

$p$CO₂ and water-air CO₂ fluxes significantly increased with greater disturbance in Australian estuaries (low to very high disturbance, n = 356, 633, 731, and 888, respectively) (Fig. 2B). For example, mean water-air CO₂ fluxes across all estuary types increased from $34.1 \pm 0.8$ mmol CO₂-C m⁻² d⁻¹ in the low disturbance systems to $63.8 \pm 1.2$ mmol CO₂-C m⁻² d⁻¹ in the very high disturbance systems (Table 1). However, the effect of disturbance on $p$CO₂ and water-air CO₂ fluxes was estuary type specific (Fig. 3). In the lagoons, $p$CO₂ in the low disturbance systems ($n = 41$) was below atmospheric equilibrium (Fig. 3A1), with CO₂ influx from the atmosphere into the estuarine waters (range: 17 to 436 μatm, −46.2 to 4 mmol CO₂-C m⁻² d⁻¹; Figure 3A2). $p$CO₂ increased significantly in higher disturbance lagoons ($p = 0.001$), except between highly and very highly disturbed lagoons ($p = 0.934$) (moderate to very high disturbance, n = 161, 261, and 288, respectively). Similarly, water-air CO₂ flux in lagoons increased significantly with higher disturbance ($p = 0.001$), but only from low to high disturbance, and was significantly lower in very high disturbance lagoons compared to high disturbance lagoons ($p = 0.037$).

In the small deltas, $p$CO₂ was significantly higher in the very high disturbance systems ($n = 366$) compared to the high disturbance systems ($n = 353$) ($p = 0.001$), but water-air CO₂ flux was similar between the high and very high disturbance systems ($p = 0.101$) (Fig. 3B). No measurements were taken in low and moderate disturbance small deltas. In the tidal systems, disturbance effects on $p$CO₂ were insignificant in the low ($n = 315$) and moderate ($n = 472$) ($p = 0.682$), and low and high ($n = 117$) disturbance systems ($p = 0.118$) but significantly increased from the moderate to high disturbance systems ($p \geq 0.006$) (Fig. 3). $p$CO₂ in the very high ($n = 234$) disturbance tidal systems was significantly greater than in low disturbance systems ($p = 0.001$). Water-air CO₂ fluxes significantly increased with higher disturbance ($p = 0.001$) except between the low and high disturbance systems ($p = 0.094$). Water-air CO₂ fluxes were greatest in very high disturbance systems (Fig. 3C2 and Table 1).

**Table 1 | Descriptive statistics calculated for $pCO_2$ and water-air $CO_2$ fluxes using data at per-minute resolution of each estuary type, disturbance group, and in the disturbance groups within each estuary type**

| | | $pCO_2$ (µatm) | | | | | | Water-air $CO_2$ flux (mmol $CO_2$-C m$^{-2}$ d$^{-1}$) | | | | | |
| | | Per-minute (averaged per-estuary) | | | | | | Per-minute (averaged per-estuary) | | | | | |
| Estuary type | Disturbance | Mean | Median | SE | IQR | Min | Max | Mean | Median | SE | IQR | Min | Max |
|---|---|---|---|---|---|---|---|---|---|---|---|---|---|
| Lagoons | All | 799 | 555 | 13 | 336 | 17 (334) | 9478 (1906) | 26.4 | 12.4 | 0.9 | 30.5 | −64.7 (−12.6) | 546.7 (85.6) |
| Small deltas | All | 1181 | 1012 | 12 | 1054 | 344 (413) | 4342 (2703) | 63.9 | 42.9 | 1.1 | 82.8 | −2.6 (2.1) | 401.1 (234.1) |
| Tidal systems | All | 1007 | 849 | 6 | 611 | 329 (585) | 2906 (1611) | 54.8 | 41.3 | 0.8 | 45.4 | −3.5 (11.8) | 570.8 (131.8) |
| All | Low | 864 | 751 | 10 | 569 | 17 (457) | 2162 (855) | 34.1 | 24.2 | 0.8 | 38.0 | −46.2 (0) | 172.5 (48.5) |
| All | Moderate | 869 | 762 | 9 | 341 | 116 (454) | 9478 (1927) | 39.7 | 36.4 | 0.6 | 39.9 | −64.7 (−5.5) | 193.8 (75.4) |
| All | High | 977 | 744 | 11 | 737 | 233 (405) | 5371 (2238) | 46.6 | 29.8 | 1.1 | 49.9 | −46 (−3) | 546.7 (183.9) |
| All | Very high | 1152 | 846 | 12 | 1058 | 199 (417) | 5791 (2454) | 63.8 | 39.1 | 1.2 | 75.9 | −9.2 (2) | 570.8 (187.8) |
| Lagoons | Low | 185 | 163 | 10 | 285 | 17 (92) | 436 (203) | −18.7 | −18.0 | 0.6 | 5.6 | −46.2 (−29) | 4 (−9.5) |
| | Moderate | 657 | 559 | 28 | 194 | 116 (378) | 9478 (1980) | 10.1 | 9.2 | 0.8 | 9.9 | −64.7 (−16.9) | 193.8 (48.3) |
| | High | 859 | 536 | 22 | 509 | 233 (337) | 5371 (2357) | 37.7 | 20.7 | 2.0 | 37.8 | −46 (−11.3) | 546.7 (177.7) |
| | Very high | 916 | 604 | 23 | 377 | 199 (403) | 5791 (2297) | 31.9 | 14.2 | 1.2 | 35.8 | −9.2 (-0.6) | 233.8 (99.7) |
| Small deltas | High | 1063 | 842 | 16 | 904 | 350 (410) | 3992 (2535) | 57.2 | 37.1 | 1.6 | 69.9 | −2.1 (0.5) | 401.1 (230.9) |
| | Very high | 1296 | 1245 | 17 | 1113 | 344 (417) | 4342 (2870) | 70.3 | 51.2 | 1.6 | 96.3 | −2.6 (3.7) | 374.5 (237.3) |
| Tidal systems | Low | 956 | 840 | 9 | 609 | 577 (731) | 2162 (1344) | 41.3 | 29.1 | 0.7 | 38.2 | 10.9 (21.7) | 172.5 (92) |
| | Moderate | 942 | 822 | 8 | 411 | 329 (562) | 2846 (1854) | 49.9 | 43.0 | 0.6 | 30.4 | −3.5 (10.6) | 164.1 (113.3) |
| | High | 981 | 934 | 17 | 647 | 406 (558) | 1683 (1052) | 34.6 | 34.2 | 1.0 | 27.4 | −0.1 (7.5) | 114.8 (58.2) |
| | Very high | 1217 | 1067 | 20 | 1017 | 397 (445) | 2906 (1937) | 92.8 | 72.2 | 3.0 | 85.1 | −2.4 (3.7) | 570.8 (264.8) |

The mean of minimum and maximum values calculated for each estuary are presented in brackets.
*SE* standard error, *IQR* interquartile range (3rd quartile–1st quartile).

## Seasonal $CO_2$ emissions from Australian estuaries

We estimated that Australian estuaries emit a mean (±SE) of 7.62 ± 0.48 Tg $CO_2$-C yr$^{-1}$ over the summer season, of which tidal systems contributed 93.4%, and lagoons and small deltas contributed 4.4% and 2.2%, respectively. To estimate winter water-air $CO_2$ fluxes, seasonal ratios from published summer and winter water-air $CO_2$ fluxes from 13 estuaries (Supplementary Table 1) were averaged to obtain the seasonal ratio (winter $CO_2$ flux: summer $CO_2$ flux) for each estuary type (Supplementary Table 2) which were then applied to summer water-air $CO_2$ fluxes from the current study. In these 13 estuaries, lagoons in winter had a lower mean $CO_2$ uptake, with seasonal ratios ranging from 0.58 to 0.69 (mean: 0.64; Supplementary Table 2). Small deltas and tidal systems in winter had higher mean water-air $CO_2$ flux rates than in summer, with seasonal ratios ranging from 0.33 to 4.71 in small deltas (mean: 1.49) and 0.43 to 2.17 in tidal systems (mean: 1.3) (Supplementary Table 2). Winter water-air $CO_2$ fluxes in Australian estuaries had a mean (±SE) flux rate of 49.7 ± 7.5 mmol $CO_2$-C m$^{-2}$ d$^{-1}$, 25.8% higher than summer flux rates (Table 2).

## Annual $CO_2$ emissions from Australian estuaries

The mean annual $CO_2$ emission from Australian estuaries was 8.67 ± 0.54 Tg $CO_2$-C yr$^{-1}$ (Table 2), with tidal systems accounting for 94.4% of annual $CO_2$ emissions, followed by small deltas at 2.5%, and lagoons at 3.1%. These proportions compared with surface areas for tidal systems, lagoons, and small deltas, representing 89.9%, 8.6%, and 1.5% of total Australian estuary surface area, respectively (Table 3). Due to the larger surface area coverage of lagoons with increased disturbance (Table 3), $CO_2$ emissions from lagoons were dominated by the higher disturbance systems. High disturbance lagoons had the greatest $CO_2$ flux rates, but very high disturbance lagoons covered a greater proportion of lagoon surface area (62%) and therefore, as a category, emitted the most $CO_2$. In contrast, lower disturbance small deltas and tidal systems covered the largest proportion of their respective estuary-type surface area and emitted the most $CO_2$ annually (Tables 2 and 3). Low and moderately disturbed tidal systems

had the greatest total emissions, driven mainly by the large surface area coverage (33% and 48%, respectively) in remote northern Australia. Very high disturbance small deltas had the highest water-air fluxes and the largest proportion of total small delta surface area (50%) and therefore, emitted the most $CO_2$ annually. Low disturbance lagoons were the only $CO_2$ sinks of all the estuary types and disturbance groups, whereas in tidal systems the very high disturbance systems emitted the least $CO_2$ annually. The low, moderate, and high disturbance small deltas emitted similarly low levels of $CO_2$ annually (Table 2). Moderately disturbed Australian estuaries had the largest $CO_2$ emissions, followed by the high, low, and very high disturbance systems (Table 2).

## Discussion

There was a strong geomorphic effect on measured $pCO_2$ and water-air $CO_2$ fluxes in Australian estuaries (Fig. 2A1 and A2), with the lagoons particularly different from the small deltas and tidal systems. Overall, lagoons had the lowest $pCO_2$ and water-air $CO_2$ fluxes of the three geomorphic types. This was likely driven by higher benthic productivity, which can result in a net autotrophic system with $CO_2$ uptake across the water-air interface[43,44] over a diurnal period[45,46]. Indeed, seagrass meadows cover an average of 18% of lagoon water areas in NSW, compared to only ~6% in small deltas and tidal systems[47]. Consistent with this, $CO_2$ undersaturation and $CO_2$ uptake have been reported in three Australian lagoons[43], as well as non-Australian marine-dominated shallow coastal systems with a large cover of benthic vegetation (e.g. refs. [44,48,49]). Freshwater input could also be a driver of $pCO_2$ and water-air $CO_2$ fluxes in estuaries; freshwater is typically supersaturated with $CO_2$ and a source of allochthonous organic matter[5,50–52] that may subsequently decompose and release $CO_2$[7,51]. However, we found poor relationships between salinity and $pCO_2$ and water-air $CO_2$ fluxes in Australian lagoons (Supplementary Fig. 2), suggesting that freshwater organic matter was not an important source of $CO_2$ in these systems. This may reflect a weak hydrological connection between lagoons and upstream rivers, which would limit

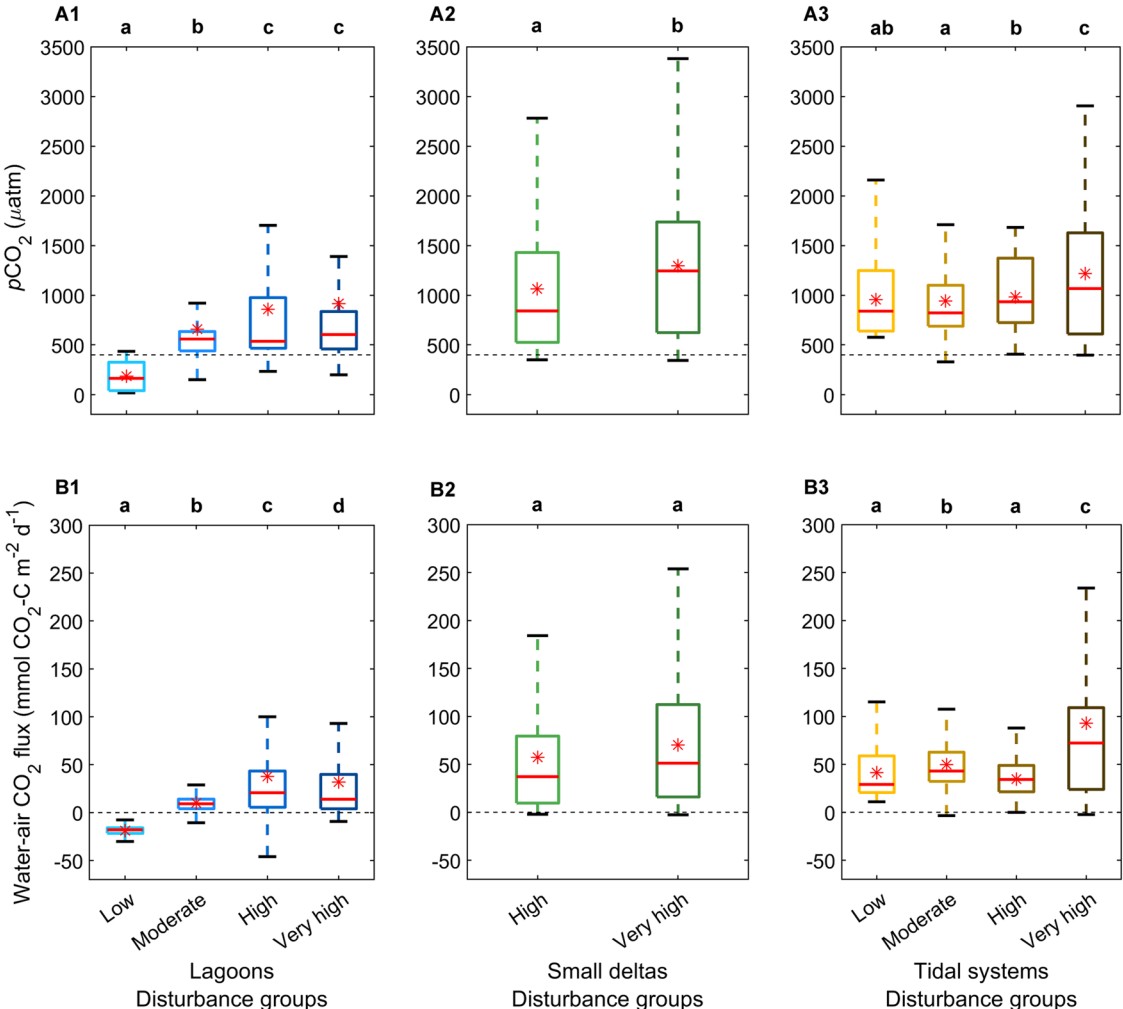

**Fig. 3 | CO₂ partial pressure (pCO₂) and water-air CO₂ flux in estuary type disturbance groups.** Median (red line), mean (red asterisk), 1st and 3rd inter-quartile ranges (box caps), minimum and maximum values (whiskers) of (row A) $pCO_2$, and (row B) water-air $CO_2$ flux at per-minute resolution across different disturbance groups within (column 1) lagoons (from low (light blue) to very high disturbance (dark blue), $n$ = 214, 815, 1312, and 1448), (column 2) small deltas ($n$; high (light green)=1777, and very high (dark green)=1845), and (column 3) tidal systems (from low (light yellow) to very high disturbance (dark brown), $n$ = 1582, 2374, 588, and 1176). Outliers were omitted from the figures. Dotted line along $x$-axis represents atmospheric $pCO_2$ and water-air $CO_2$ flux equilibrium. Letters above figures denote statistical differences among estuary types, with letters that are the same indicating no significant difference (PERMANOVA, two-tailed, and at 95% confidence interval). Source data are provided in the Source Data file.

input of riverine water. This is consistent with a previous study showing lower CO₂ emissions in estuaries with lower riverine input compared to river-dominated estuaries[53].

$pCO_2$ and DIC concentration were higher in small deltas and tidal systems compared to lagoons (Figure 2A1, Supplementary Fig. 4B1, Supplementary Results). The inverse relationships between salinity and $pCO_2$ or water-air $CO_2$ fluxes in small deltas and tidal systems indicate that CO₂ outgassing from the re-mineralisation of organic matter in upstream waters had a larger contribution in these systems compared to the lagoons[31], as seen in other estuaries with higher riverine input[53]. This is because increasing $pCO_2$ and water-air $CO_2$ fluxes with decreasing salinity indicate that increasing $CO_2$ is linked to freshwater input upstream. The tidal systems and small deltas also have a stronger connection to the river and associated input of CO₂ supersaturated water[2], which enhances CO₂ emissions in the estuary. In our study, all of the small deltas and tidal systems were tropical and sub-tropical (23.5° to 35° latitude) (Supplementary Data 1), where much of the atmospheric carbon uptake and sequestration occurs within their mangrove-lined shorelines[54,55]. Lateral export from vegetated shorelines into adjacent estuaries can be a significant pathway for the transport of carbon in the form of DOC, POC, DIC, and CO₂-rich pore water, or as a result of the degradation of exported organic matter[56–58]. Increased tidal range resulted in an increase in $pCO_2$ and water-air $CO_2$ fluxes (Supplementary Results), suggesting increased $pCO_2$ and water-air $CO_2$ fluxes were due to lateral export in our estuaries[56–58], although we did not directly measure lateral inputs. Although estuaries in lower latitudes have higher water temperatures that could drive increased water-air $CO_2$ fluxes, water temperature did not correlate with $pCO_2$ and water-air fluxes in our estuaries (Supplementary Results). As such, higher $pCO_2$ and DIC concentrations in small deltas and tidal systems compared to lagoons could likely be attributed to increased lateral inorganic (and organic) carbon export from intertidal coastal wetlands due to stronger lateral exchange by tides compared to lagoons (Fig. 1E).

In contrast to the lagoons, where DIC was positively correlated with $pCO_2$ and water-air $CO_2$ fluxes, in tidal systems and small deltas $pCO_2$ and $CO_2$ fluxes were not strongly associated with DIC concentrations; only CO₂ fluxes in tidal systems showed a very weak trend with DIC concentration (Supplementary Results). Removing the effect of salinity in our analysis (as a co-variate), the differences in $pCO_2$ and

**Table 2 | Mean, standard error (SE), and median for summer, winter, and annual water-air $CO_2$ flux rates and annual $CO_2$ emissions from Australian estuaries (Estuary (Est.) types: La: lagoons, SD: small deltas, TS: tidal systems; Disturbance (Dist.) groups: 1: low, 2: moderate, 3: high, 4: very high)**

| Est. type | Dist. | Summer Water-air $CO_2$ flux (mmol $CO_2$-C $m^{-2}$ $d^{-1}$) Median | Mean | SE | Winter Water-air $CO_2$ flux (mmol $CO_2$-C $m^{-2}$ $d^{-1}$) Median | Mean | SE | Mean (−) | Mean (+) | Annual (Summer + Winter) Water-air $CO_2$ flux (mmol $CO_2$-C $m^{-2}$ $d^{-1}$) Median | Mean | SE | Mean (−) | Mean (+) | Australian $CO_2$ emissions (Tg $CO_2$-C $yr^{-1}$) Median | Mean | SE | Mean (−) | Mean (+) |
|---|---|---|---|---|---|---|---|---|---|---|---|---|---|---|---|---|---|---|---|
| La | 1 | −18.3 | −18.8 | 0.9 | −11.6 | −12.0 | 0.6 | −11.0 | −13.0 | −15.0 | −15.4 | 0.8 | −14.9 | −15.9 | −0.02 | −0.02 | 0.00 | −0.02 | −0.02 |
|  | 2 | 9.9 | 12.5 | 5.2 | 6.3 | 7.9 | 3.3 | 7.3 | 8.6 | 8.1 | 10.2 | 4.3 | 9.9 | 10.5 | 0.01 | 0.01 | 0.01 | 0.01 | 0.01 |
|  | 3 | 19.5 | 30.4 | 10.8 | 12.4 | 19.3 | 6.9 | 17.7 | 20.9 | 16.0 | 24.8 | 8.8 | 24.0 | 25.6 | 0.05 | 0.08 | 0.03 | 0.07 | 0.08 |
|  | 4 | 21.1 | 26.8 | 7.4 | 13.4 | 17.0 | 4.7 | 15.6 | 18.5 | 17.2 | 21.9 | 6.1 | 21.2 | 22.6 | 0.16 | 0.20 | 0.06 | 0.19 | 0.21 |
|  | All | 12.8 | 16.3 | 5.0 | 8.1 | 10.4 | 3.2 | 9.5 | 11.3 | 10.5 | 13.4 | 4.1 | 12.9 | 13.8 | 0.20 | 0.27 | 0.05 | 0.26 | 0.28 |
| SD | 1[a] | – | – | – | – | – | – | – | – | – | – | – | – | – | 0.03 | 0.03 | – | 0.02 | 0.08 |
|  | 2[a] | – | – | – | – | – | – | – | – | – | – | – | – | – | 0.03 | 0.03 | – | 0.02 | 0.07 |
|  | 3 | 51.4 | 58.3 | 14.7 | 76.8 | 87.1 | 22.0 | 19.2 | 275.0 | 64.1 | 72.7 | 18.3 | 38.8 | 166.7 | 0.03 | 0.04 | 0.01 | 0.02 | 0.08 |
|  | 4 | 61.9 | 70.2 | 10.7 | 92.5 | 104.8 | 15.9 | 23.1 | 330.7 | 77.2 | 87.5 | 13.3 | 46.6 | 200.4 | 0.10 | 0.11 | 0.02 | 0.06 | 0.26 |
|  | All | 58.8 | 64.2 | 8.8 | 87.9 | 95.9 | 13.2 | 21.2 | 302.9 | 73.3 | 80.1 | 11.0 | 42.7 | 183.6 | 0.19 | 0.21 | 0.03 | 0.11 | 0.49 |
| TS | 1 | 44.1 | 46.1 | 12.4 | 57.4 | 60.0 | 16.1 | 19.7 | 100.3 | 50.7 | 53.1 | 14.2 | 32.9 | 73.2 | 2.58 | 2.70 | 0.72 | 1.67 | 3.72 |
|  | 2 | 50.0 | 51.8 | 4.0 | 65.1 | 67.4 | 5.2 | 22.2 | 112.7 | 57.6 | 59.6 | 4.6 | 37.0 | 82.3 | 4.33 | 4.48 | 0.35 | 2.78 | 6.19 |
|  | 3 | 22.2 | 22.2 | 21.8 | 28.9 | 28.9 | 28.4 | 9.5 | 48.3 | 25.6 | 25.6 | 25.1 | 15.9 | 35.3 | 0.63 | 0.63 | 0.62 | 0.39 | 0.87 |
|  | 4 | 72.1 | 84.4 | 42.6 | 93.8 | 109.8 | 55.4 | 36.1 | 183.5 | 82.9 | 97.1 | 49.0 | 60.2 | 134.0 | 0.21 | 0.25 | 0.13 | 0.15 | 0.34 |
|  | 0[b] | – | – | – | – | – | – | – | – | – | – | – | – | – | 0.11 | 0.12 | 0.02 | 0.08 | 0.17 |
|  | All | 49.1 | 52.9 | 10.2 | 63.9 | 68.9 | 13.3 | 22.6 | 115.1 | 56.5 | 60.9 | 11.8 | 37.8 | 84.0 | 7.86 | 8.18 | 0.91 | 5.08 | 11.29 |
| All est. |  | 36.6 | 39.5 | 5.3 | 35.3 | 49.7 | 7.5 | 16.4 | 116.7 | 31.2 | 44.6 | 6.4 | 27.9 | 78.1 | 8.25 | 8.67 | 0.54 | 5.45 | 12.06 |
| All | 1 | 23.3 | 18.3 | 14.7 | 30.3 | 29.2 | 16.9 | 6.6 | 51.8 | 26.8 | 23.7 | 15.8 | 12.4 | 35.0 | 1.41 | 1.25 | 0.83 | 0.65 | 1.84 |
| All | 2 | 29.6 | 28.9 | 6.7 | 18.8 | 32.7 | 9.3 | 13.5 | 52.0 | 24.2 | 30.8 | 7.9 | 21.2 | 40.4 | 1.86 | 2.37 | 0.61 | 1.63 | 3.11 |
| All | 3 | 36.6 | 42.0 | 9.1 | 44.3 | 52.1 | 14.1 | 17.1 | 142.4 | 40.2 | 47.1 | 11.4 | 29.6 | 92.2 | 1.14 | 1.33 | 0.32 | 0.84 | 2.61 |
| All | 4 | 51.0 | 55.7 | 10.8 | 65.0 | 70.7 | 16.3 | 22.7 | 176.4 | 54.3 | 63.2 | 13.5 | 39.2 | 116.0 | 0.70 | 0.82 | 0.17 | 0.51 | 1.51 |

Descriptive statistics for summer water-air $CO_2$ fluxes were calculated with per-minute resolution data measured in this study while winter water-air fluxes were calculated based on seasonal ratios. Up-adjusted means (+) (using maximum seasonal ratios) and down-adjusted means (−) (using minimum seasonal ratios) calculated from the sensitivity analysis are shown for winter and annual water-air $CO_2$ fluxes and annual $CO_2$ emissions.

[a]Mean and median low and moderate disturbance small delta annual $CO_2$ emissions were calculated using mean and median small delta water-air $CO_2$ flux rates.

[b]Mean and median annual $CO_2$ emissions from tidal systems with disturbance classified no assessment were calculated using mean and median tidal system water-air $CO_2$ flux rates.

**Table 3 | Number of estuaries, estuarine surface area, and percent of total estuary surface area classified by estuary type according to Dürr et al.[2] and classified by disturbance in NLWRA[40] for sampled estuaries in this study and in all Australia**

| Estuary type | Disturbance | Study surface area coverage | | | National surface area coverage | | |
|---|---|---|---|---|---|---|---|
| | | Estuaries (n) | (km²) | % National representation | Estuaries (n) | (km²) | % Estuary type |
| Lagoons | Low | 3 | 38 | 13.3 | 78 | 286 | 8.4 |
| | Moderate | 7 | 95 | 31.0 | 75 | 308 | 9.1 |
| | High | 5 | 226 | 32.1 | 82 | 704 | 20.8 |
| | Very high | 6 | 286 | 13.7 | 36 | 2083 | 61.6 |
| | Not assessed | 0 | 0 | – | 2 | 0 | – |
| | Total | 21 | 645 | 19.1 | 273 | 3382 | 8.6 |
| Small deltas | Low | 0 | 0 | 0.0 | 38 | 99 | 16.7 |
| | Moderate | 0 | 0 | 0.0 | 39 | 85 | 14.4 |
| | High | 6 | 18 | 16.3 | 47 | 112 | 18.9 |
| | Very high | 6 | 100 | 34.0 | 25 | 295 | 49.9 |
| | Not assessed | – | – | – | 0 | 0 | – |
| | Total | 12 | 119 | 20.1 | 149 | 591 | 1.5 |
| Tidal systems | Low | 4 | 1012 | 8.7 | 359 | 11598 | 32.7 |
| | Moderate | 5 | 1350 | 11.6 | 97 | 17152 | 48.4 |
| | High | 2 | 1553 | 13.4 | 61 | 5630 | 15.9 |
| | Very high | 3 | 179 | 1.5 | 27 | 582 | 1.6 |
| | Not assessed | 0 | 0 | – | 5 | 455 | 1.3 |
| | Total | 14 | 4094 | 11.6 | 549 | 35417 | 89.9 |
| Total | | 47 | 4858 | 12.3[a] | 971 | 39390 | 100 |
| Disturbance | Estuary type | Estuaries (n) | (km²) | % National representation | Estuaries (n) | (km²) | % Disturbance group |
| Low | Lagoons | | | 0.3 | | | 2.4 |
| | Small deltas | | | 0 | | | 0.8 |
| | Tidal systems | | | 8.4 | | | 96.8 |
| | Total | 7 | 1050 | 8.8 | 475 | 11983 | 30.4 |
| Moderate | Lagoons | | | 0.5 | | | 1.8 |
| | Small deltas | | | 0 | | | 0.5 |
| | Tidal systems | | | 7.7 | | | 97.8 |
| | Total | 12 | 1446 | 8.2 | 211 | 17545 | 44.5 |
| High | Lagoons | | | 3.5 | | | 10.9 |
| | Small deltas | | | 0.3 | | | 1.7 |
| | Tidal systems | | | 24.1 | | | 87.3 |
| | Total | 13 | 1797 | 27.9 | 190 | 6446 | 16.4 |
| Very high | Lagoons | | | 9.7 | | | 70.4 |
| | Small deltas | | | 3.4 | | | 10.0 |
| | Tidal systems | | | 6.0 | | | 19.7 |
| | Total | 15 | 565 | 19.1 | 88 | 2961 | 7.5 |
| Not assessed | Lagoons | | | – | | | 0.1 |
| | Small deltas | | | – | | | 0 |
| | Tidal systems | | | 0 | | | 99.9 |
| | Total | – | – | – | 7 | 455 | 1.2 |
| Total | | 47 | 4858 | 12.3 | 971 | 39390 | 100 |

Two lagoons and five tidal systems were not assessed for disturbance[40].
[a]Total represented coverage of the national estuarine surface area.

water-air $CO_2$ flux correlations with DIC between estuary types suggests that other factors specific to small deltas and tidal systems further influence $CO_2$ and DIC dynamics along the estuarine gradient in those systems. For example, shorter water residence times and increased intertidal wetlands are among factors driving processes impacting $CO_2$ and DIC, which may include $CO_2$ emissions of mangrove porewater DIC and mineralisation of exported POC and DOC[55,59,60]. The positive relationship between $p CO_2$ and DOC (Supplementary Results and Supplementary Table 5) suggests that DOC

mineralisation may drive increased $p CO_2$ in small deltas. In tidal systems, stronger tidal influence compared to small deltas could further promote $CO_2$ emissions through the tidal resuspension of sediments, releasing DIC and organic matter for remineralisation[61–63], with excess DIC and organic matter exported to the coastal ocean[58,64].

The sensitivity analysis showed that summer:winter water-air $CO_2$ flux ratios ranged from 0.33 to 4.71 across the 13 estuaries (Supplementary Table 1). The largest range in summer:winter water-air flux $CO_2$ ratios was for small deltas, where ratios were up to 3x larger than

in lagoons and tidal systems (Supplementary Table 2). However, when estimating annual emissions, the large range in ratios in small deltas was attenuated by the large surface area of tidal systems compared to the small surface area coverage by small deltas (Table 3). Including mean winter water-air flux rates in annual Australian estuarine $CO_2$ emission calculations only showed 13% greater water-air flux rate and 14% greater annual emissions than from summer measurements alone. Although our study accounts for $pCO_2$ variations in seasonality extremes, it does not account for variations due to diurnal cycles and episodic events such as flooding. However, the diurnal effect on $CO_2$ emissions from estuarine surface waters is likely minimal, with differences between day and night driven more by tidal influence[65–67]. Episodic events are more significant drivers of increased $CO_2$ emissions in estuarine surface waters[68], but quantifying the effect of these events on $CO_2$ emissions in Australian estuaries was beyond the scope of this study.

The seasonal variability of water-air $CO_2$ fluxes in Australian estuaries is consistent with other studies globally, showing a range of change between seasons. For example, mean $CO_2$ water-air fluxes were highest in autumn (36.2 mmol C m$^{-2}$ d$^{-1}$) (mean water temperature: 11.5 °C) followed by spring (24.1 mmol C m$^{-2}$ d$^{-1}$) (7.5 °C) and summer (18.2 mmol C m$^{-2}$ d$^{-1}$) (17.5 °C), and lowest in winter (7.9 mmol C m$^{-2}$ d$^{-1}$) (2.9 °C) in the temperate Tay estuary (tidal system) in the United Kindom[69]. In contrast, significantly higher mean water-air $CO_2$ fluxes were found in winter (15.6 ± 5.2 mmol C m$^{-2}$ d$^{-1}$) and summer (13.4 ± 22.2 mmol C m$^{-2}$ d$^{-1}$, highest discharge rate) than in spring (−13.7 ± 16.4 mmol C m$^{-2}$ d$^{-1}$) and autumn (2.7 ± 6.6 mmol C m$^{-2}$ d$^{-1}$) in the Delaware Estuary, USA (tidal system) (2015 water temperature range: 0.4 °C to 28.6 °C[70], temperature data: https://waterdata.usgs. gov/monitoring-location/01463500/#parameterCode= 00010&startDT=2015-01-01&endDT=2016-01-01). In the current study, there was no correlation between water-air $CO_2$ fluxes and temperature across all estuaries and within estuary types (studied estuary temperature range: 16 °C to 34.3 °C), except for a weak, significant correlation in tidal systems. This suggests that water-air $CO_2$ fluxes in our study were likely driven by factors other than temperature, for example, riverine and lateral inputs into the estuaries or residence times.

Importantly, the seasonal variability between summer and winter water-air $CO_2$ flux rates was small compared to the variability within individual estuaries and estuary types. Variability within the estuary types ranged from a maximum per-minute water-air $CO_2$ flux rate 10 times (9 times as an estuary average minimum) larger than the minimum rate in the lagoons, 156 (113) times larger in the small deltas, and 165 (13) times larger in the tidal systems (Table 1). This larger within-estuary type spatial variability in water-air $CO_2$ flux rates was captured in our continuous sampling along each estuary. We also accounted for the likely range of seasonal variability in water-air $CO_2$ flux rates (maximum in summer, minimum in winter) by applying a summer:winter ratio to our summer data. As such, we argue that our annual emissions estimates are fairly robust.

Lagoons had the strongest disturbance signal, with $pCO_2$ and water-air $CO_2$ fluxes increasing with increasing disturbance (Figure 3A1 and B1), mainly driven by changes in the extent of seagrass cover. With increasing disturbance, NSW lagoons had a general decrease in seagrass cover (low = 55%, moderate = 19%, high = 24%, and very high = 3%)[47] and mean dissolved oxygen (low = 123%$_{sat}$, moderate = 97%$_{sat}$, high = 95%$_{sat}$, and very high = 108%$_{sat}$) (Supplementary Fig. 4C2). Despite relying on data that was over 15 years old (mapped in 2007–2009)[71,72], percent seagrass cover had a strong, negative association with $pCO_2$ and a weaker, negative association with water-air $CO_2$ fluxes (Supplementary Fig. 5 and Supplementary Results). These relationships suggest that higher $pCO_2$ and water-air $CO_2$ fluxes reflect decreased $CO_2$ uptake by benthic vegetation (i.e. a less autotrophic system), as reported in several seagrass studies (e.g. refs. 43,44,73). High DOC concentrations in the low-disturbance

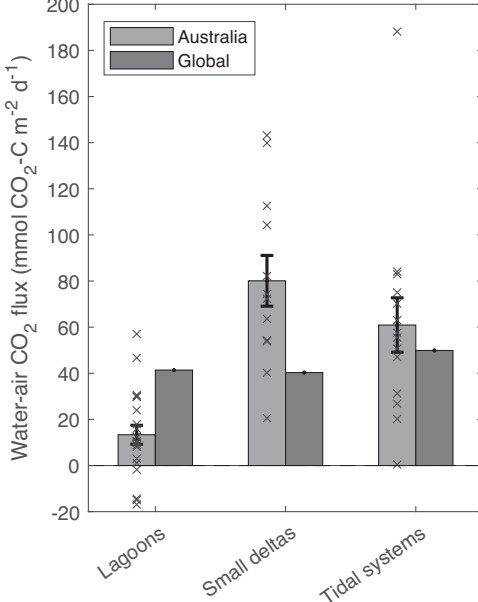

**Fig. 4 | Australian and global estuarine $CO_2$ emissions by estuary type.** Mean water-air $CO_2$ fluxes in Australian estuaries ($n$: Lagoons = 21, Small deltas = 12, and Tidal systems = 14) and in global estuaries[10] from the three estuary types defined in this study. Error bars represent standard errors. Source data are provided as a Source Data file.

lagoons were also consistent with DOC release from benthic vegetation, as observed in previous studies[48,74,75] (Supplementary Fig. 4A2, Supplementary Table 5, and Supplementary Results). High percent $O_2$ saturation in the very high disturbance lagoons (e.g. Curl Curl Lagoon; Supplementary Results, Supplementary Data 2, and Supplementary Table 4) was most likely due to a switch in production from benthic microalgae and macroalgae to phytoplankton[76], enabling enhanced $pCO_2$ drawdown and negative water-air $CO_2$ flux rates (Figs. 3A1, B1).

This study estimates that Australia's estuaries have a mean (±SE) annual area-weighted water-air $CO_2$ emissions of 44.6 ± 6.4 mmol $CO_2$-C m$^{-2}$ d$^{-1}$, which is 25% and 110% greater, respectively, than estimates of global means of 35.6 mmol $CO_2$-C m$^{-2}$ d$^{-1}$ [10] and 21.2 mmol $CO_2$-C m$^{-2}$ d$^{-1}$ [9]. However, the role of estuary type in $CO_2$ flux rates had a significant impact on our estimates of Australian water-air $CO_2$ flux. Annual mean (±SE) area-weighted water-air $CO_2$ fluxes of the lagoons upscaled to all Australian lagoons (13.4 ± 4.1 mmol $CO_2$-C m$^{-2}$ d$^{-1}$) was 68% lower than from global lagoon estimates (41.4 mmol $CO_2$-C m$^{-2}$ d$^{-1}$)[10]. In contrast, annual mean water-air $CO_2$ fluxes (Table 2 and Fig. 4) scaled to all Australian small deltas (80.1 ± 11 mmol $CO_2$-C m$^{-2}$ d$^{-1}$) and tidal systems (60.9 ± 11.8 mmol $CO_2$-C m$^{-2}$ d$^{-1}$) were 99% and 22% higher, respectively, than from global small deltas (40.3 mmol $CO_2$-C m$^{-2}$ d$^{-1}$) and tidal systems (49.9 mmol $CO_2$-C m$^{-2}$ d$^{-1}$)[10]. In addition to the higher water-air $CO_2$ fluxes in Australian small deltas and tidal systems, the lower global mean water-air $CO_2$ flux for all estuaries combined compared to Australia likely also reflects the contribution of fjords and fjards to global estimates[9,10]. Globally, fjords and fjards have been shown to take up $CO_2$ from the atmosphere (median 66 Tg $CO_2$ yr$^{-1}$)[11], but are absent in Australia.

Lower mean $CO_2$ emissions in Australian lagoons compared to lagoons globally are likely due to overall lower disturbance in Australia. In addition, it may also reflect the greater abundance of ICOLLs in Australia (21% of global occurrence[41,77,78]). Isolation from marine waters, low riverine flow, and long residence times in ICOLLs may enhance autotrophic drawdown of $CO_2$ by abundant seagrasses,

resulting in smaller water-air $CO_2$ fluxes (Table 2) than observed in non-Australian lagoons[43,79,80]. The weak hydrological connection between ICOLLs and rivers would also limit the input of $CO_2$ supersaturated river water.

Higher mean water-air $CO_2$ fluxes in Australian subtropical and tropical small deltas and tidal systems, compared to small deltas and tidal systems globally was an unexpected finding. Subtropical and tropical estuaries have previously been estimated to have lower water-air $CO_2$ fluxes than systems at temperate latitudes[9]. We were unable to explicitly test for climate as we do not have sufficient estuaries of each geomorphic type and disturbance in each climate zone. However, macrotidal northern Australian small deltas and tidal systems are different from the small deltas and tidal systems in previous studies[81] (Fig. 6 in Matthews and Matthews[82]) as North Australian tidal systems are dominated by extensive mangrove cover and have significantly greater tidal ranges (>4 m). Larger tidal ranges would lead to greater lateral inorganic and organic carbon export from mangroves to the tidal systems[56,58,83]. Higher mean water-air $CO_2$ fluxes may also reflect the longer residence times resulting from characteristically low Australian freshwater inflows[84,85]. Long residence times would allow more time for $CO_2$ produced from lateral inputs of DIC, DOC, and POC, and DIC from increased DOC and POC decomposition to be emitted across the water-air interface rather than flushed to the ocean.

Collectively, estuary geomorphic type is more important than disturbance in Australia, resulting in higher mean $CO_2$ emissions from Australian estuaries despite their lower overall disturbance compared to global estuaries. The climate zone also has an important control on estuarine geomorphic type (e.g. tropical and subtropical mangrove-dominated macrotidal estuaries). This study suggests that relative to their surface area, Australian estuaries contribute a disproportionately large amount of $CO_2$ emissions annually to global estuarine emissions. Using surface area estimates for Australian (62,100 km$^2$; calculated from Table 3 in Chen et al.[9]) and global estuaries (1,012,440 km$^2$) and global estuarine $CO_2$ emission estimates by Laruelle et al.[10] and Chen et al.[9], Australian estuaries emit a mean ($\pm$SE) of 12.1 $\pm$ 1.7 Tg $CO_2$-C annually. These emissions account for 12% or 8% of the estimated mean global estuarine $CO_2$ emissions of 0.1 Pg $CO_2$-C yr$^{-1}$[9] or 0.15 Pg $CO_2$-C yr$^{-1}$[10], despite Australian estuaries accounting for only 6.1% of their calculated global estuarine surface area. This estimate includes the surface area coverage of the estuary types and disturbance groups and is dependent on the accuracy of surface areas for Australian and global estuaries. For instance, total estuarine water area of 39,390 km$^2$ has been reported for Australia (Table 3)[40], which is 57% greater than estimated by Dürr et al.[2] (25,056 km$^2$) and 37% smaller than estimated by Laruelle et al.[10] (62,100 km$^2$). Applying the estuarine surface areas of Australia's National Land and Water Resources Audit (NLWRA)[40] to data collected in the current study, Australian estuaries are estimated to emit (mean $\pm$ SE) 8.67 $\pm$ 0.54 Tg $CO_2$-C annually (Table 2).

Australian tidal systems contributed the majority of the mean ($\pm$SE) annual $CO_2$ emissions (8.18 $\pm$ 0.91 Tg $CO_2$-C yr$^{-1}$, 94.4%), with far smaller contributions from lagoons and small deltas (Table 2). Although lagoons in Australia (8.6% of total area) cover six times the estuarine surface area of small deltas (1.5% of total area;), $CO_2$ emissions from lagoons were disproportionately low (0.27 $\pm$ 0.05 Tg $CO_2$-C yr$^{-1}$, 3.1% of Australian estuarine emissions) compared to small deltas (0.21 $\pm$ 0.03 Tg $CO_2$-C yr$^{-1}$, 2.5%) reflecting smaller water-air fluxes in lagoons (Table 2). The proportions of $CO_2$ emitted by the different geomorphic types of estuaries in Australia were different from the proportions reported globally. For example, lagoons globally account for a larger proportion of $CO_2$ emissions (31%; 0.046 Pg $CO_2$-C yr$^{-1}$) than small deltas (13%; 0.019 Pg $CO_2$-C yr$^{-1}$), and tidal systems only contribute 41% of global emissions (0.063 Pg $CO_2$-C yr$^{-1}$)[10]. The remaining proportion is made up of limited or non-filtering estuary types such as large rivers, karst-dominated coasts, and arheic coasts[2]. Furthermore, global $CO_2$ emission estimates incorporate the

contribution of fjords and fjards, which have the lowest water-air $CO_2$ flux or show $CO_2$ uptake but account for close to half of the global estuarine surface area[10,11]. Therefore, differences in Australian and global estuarine $CO_2$ emissions are driven mostly by the geomorphic type (related to the climate zone of the estuary). This highlights the need to include geomorphic types in global $CO_2$ emission assessments.

Geomorphology and disturbance influence water-air $CO_2$ fluxes in Australian estuaries as a result of decreased hydrological connectivity in lagoons, and increased upstream riverine lateral inputs and tidal influence in small deltas and tidal systems. Water-air $CO_2$ flux rates increase with higher disturbance, but geomorphology and disturbance interact, with the strongest disturbance signal in the lagoons, and a weak disturbance signal in the small deltas and tidal systems. Seasonal variations in $CO_2$ emissions were a less important control on water-air $CO_2$ fluxes in Australian estuaries. Previous global estuarine $CO_2$ emission estimates have included geomorphology[9–11], but not disturbance or the two factors together. $CO_2$ emissions for global lagoons could therefore be over-estimated due to the bias towards more disturbed systems in the northern hemisphere. In contrast, $CO_2$ emissions for global small deltas and tidal systems could be under-estimated due to the bias towards temperate systems in the northern hemisphere. As such, upscaling of global estuarine $CO_2$ emissions should be based on geomorphic estuary-types but also consider land-use disturbance and climate and ideally, their interaction with geomorphic type.

## Methods

In 36 estuaries around Australia, $pCO_2$, DIC, DOC, physicochemistry, and physical parameters (wind speed, depth, current velocity, and barometric pressure) were measured along the salinity gradient from the marine to freshwater endmember (where possible). Data were combined with published $CO_2$ fluxes and water quality data for 11 other Australian estuaries[12,30], giving a total of 47 Australian estuaries. The same survey methods were used in all 47 estuaries. $pCO_2$ and water-air $CO_2$ fluxes were classified according to estuary type (lagoons, small deltas, and tidal systems) and disturbance group (low, moderate, high, and very high) and analysed for significant differences. Finally, the classified water-air $CO_2$ fluxes were upscaled to all of Australia and mean estimates were compared to previous global mean estimates of estuary $CO_2$ emissions. While we provide a full set of statistics in this study, we use the mean ($\pm$SE) for global comparison because our high-resolution water-air $CO_2$ fluxes over a range of disturbance classes and geomorphic estuary types was better represented by the means than medians.

### Estuary classification schemes

Estuaries were selected to cover a large range of disturbance and geomorphic types according to the classifications of NLWRA[40] and Dürr et al.[2]. NLWRA[40] assessed 971 Australian estuaries and described four disturbance classes (low (near-pristine), moderate (relatively unmodified), high (modified), and very high (extensively modified)). These disturbance groups were qualitatively classified based on changes in catchment land-use, estuary use, and ecology (Supplementary Table 6) and provided an assessment that was more relevant than adopting a single set of indicators. This is because the Australian continent covers a large surface area, encompassing over 1000 estuaries and climatic variations, making a single set of disturbance indicators likely misleading[86,87]. The global estuarine typology of Dürr et al.[2] details three geomorphic types found in Australia: (1) lagoons (including Intermittently Closed or Open Lakes and Lagoons (ICOLLs) and estuaries with a central basin morphology), (2) small deltas, and (3) tidal systems (including drowned river valleys and tidal embayments), based around morphological and sedimentation characteristics driven by tidal influence (classification criteria in Supplementary Table 7). However, the existing classification of Australian estuaries[2] did not match our observations of satellite imagery, because it was developed

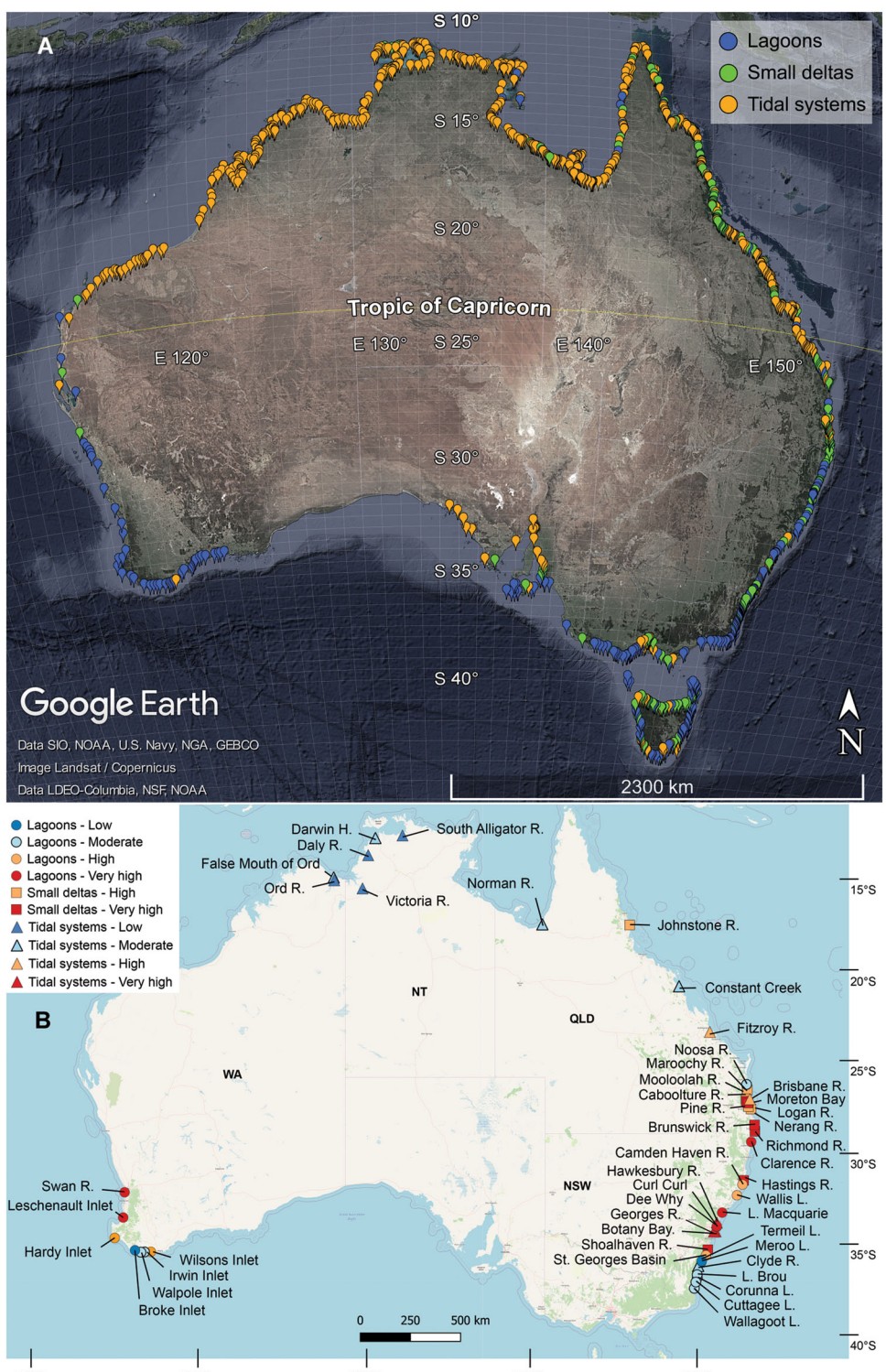

**Fig. 5 | Distribution of estuary types and study locations in Australia.**
**A** Estuaries in Australia[40] classified into three estuary types based on conceptual definitions (Supplementary Table 7) by Dürr et al.[2] (©Google Earth) and the (**B**) location of study estuaries in Australia according to estuary type (shapes) and disturbance class (colours) (©OpenStreetMap, https://www.openstreetmap.org/copyright). H.: Harbour, R.: River, and L.: Lake.

with a low spatial resolution (0.5°, or 50 km). Therefore, all 971 Australian estuaries were re-classified into the three estuary types by distinguishing physical characteristics based on the criteria of Dürr et al.[2] (Supplementary Table 7) using satellite imagery (Google Earth) (Supplementary Data 3). Water surface area for 108 estuaries with missing surface area measurements were also calculated using satellite imagery (Google Earth). The re-classified estuary database was then combined with the estuarine disturbance database in NLWRA[40] (dataset URL: https://data.gov.au/dataset/ds-aodn-8fec03d6-48e3-4352-9ddb-085e42e55637/details?q=, Supplementary Data 3).

The spatial distribution of estuary types in Australia corresponds to the tidal ranges of their respective coastlines (Fig. 5A). Tidal systems

dominate the macro-tidal regions of northern Australia, whereas lagoons are found mostly in the micro-tidal regions of southern Australia. All three estuary types with all four disturbance groups, except for low and moderate disturbance in small deltas, were included in our estuary selection (Table 3). The surface area of estuaries sampled and included in this study represents 12.3% of the total Australian estuarine surface area, consisting of 19.1% of lagoons, 20.1% of small deltas, and 11.6% of tidal systems in Australia (Table 3).

## Study sites
Measurements from the 36 estuary surveys and from the 11 published estuary surveys were taken over the austral spring-summer season (Fig. 5B and Supplementary Data 1). The estuaries included in this study were comprised of 21 estuaries in New South Wales (Nov to Dec 2017), one in southeast Queensland (Moreton Bay, Oct 2018), seven along the north Australian coastline (from Karumba, Queensland to Wyndham, Western Australia, Oct to Dec 2018), seven along the southwest coastline of Western Australia (from Albany to Perth, Feb to Mar 2019), three in north-east Queensland[30] (late spring Sept to Oct 2014) and eight in southeast Queensland[12] (late spring Oct 2016) (Fig. 5B and Supplementary Data 1). Percent seagrass coverage for the New South Wales estuaries was obtained from Roper et al.[47]. Termeil Lake and Lake Brou were excluded from seagrass coverage analysis because although zero coverage was recorded by Roper et al.[47], extensive seagrass cover was observed during our surveys.

## Underway data measurements
Using a 6 m research vessel, physicochemical parameters and $p$CO$_2$ were measured along a transect encompassing the length of each of the 36 estuaries, starting at the river mouth just after high-tide and ending in freshwater (salinity ~2). Although we aimed to finish the surveys at salinity ~2, this was not always possible because of shallow water and/or natural and artificial obstacles. As such, estuary data in this study reflect the spatial variations along the estuarine gradient (marine to upstream-riverine). A cruising ground speed of ~8 km h$^{-1}$ was maintained where possible to ensure spatial and temporal consistency. The surveys were carried out during daylight hours, typically lasting over the course of a day. Surveys in large estuaries often required 2 to 3 days but never exceeded five days (Supplementary Data 1). $p$CO$_2$ was recorded at one-minute intervals using an integrated water-gas loop setup (Supplementary Fig. 6).

Water was continuously pumped from beneath the hull (0.5 m to 1 m water depth) at ~1800 l$^{-1}$ h$^{-1}$ using a 12 V pump with backflow prevention (800GPH, Rule) to a high-flow filter basket (Ozito) before entering a two-way split. One path led to a flow-through chamber with a multi-parameter sonde (HL4, Hydrolab) measuring salinity (±0.5%), temperature (±0.1 °C), pH$_{NBS}$ (±0.2), and dissolved oxygen (DO$_{\%sat}$; ±2%). The second path entered a loop consisting of a pair of interconnected showerhead exchangers (RAD Aqua, Durridge) equilibrating dissolved gases in the water with the headspace. The dried gas stream was then measured for CO$_2$ concentration, using a LiCOR 840 A CO$_2$ gas analyser (accuracy <1%) and a Picarro G-2508 Cavity Ring-Down Spectrometer (CRDS, ±0.05%)[87]. In the ICOLL lagoons (indicated in Supplementary Data 1) where a smaller boat was used, CO$_2$ measurements were only taken with the LiCOR 840 A CO$_2$ gas analyser. Measured CO$_2$ was humidity-corrected and in-situ $p$CO$_2$ was calculated using methods in Pierrot et al.[88]. The LiCOR 840 A was calibrated using a two-step process with low (250 ppm) and high (8000 ppm) $p$CO$_2$ gas standards. The CRDS was serviced and calibrated by the manufacturer (Picarro, USA) before each field trip.

## Discrete water samples, morphological, and meteorological data
Water samples were collected for DIC and DOC concentrations, along with estuarine (depth and water current velocity) and meteorological

measurements at the start and end of survey transects and at salinity intervals of ~5. In cases where salinity did not change much (<5) along the survey, samples were collected every hour instead (i.e. every 8 km of estuary travelled). In the ICOLL lagoons (as indicated in Supplementary Data 1) where no significant salinity gradient was present, discrete water samples were collected from at least 3 points across the estuary. For DOC, 30 ml water samples were filtered through a pre-combusted (500 °C, ~5 h) 0.7 μm GF/F filter (Whatman, Merck) into an acid-washed glass vial containing 100 μl of 85% phosphoric acid (H$_3$PO$_4$). 50 ml water samples for DIC analysis were syringe-filtered (0.45 μm SFCA Minisart, Sartorius) into a crimp-top glass bottle without any headspace and preserved with 30 μl mercuric chloride (HgCl$_2$). DOC concentrations were determined using a total organic carbon analyser (±3%; 1030 W, Aurora)[89]. DIC concentrations were analysed with a Marianda AIRICA coupled to a CO$_2$/H$_2$O analyser (LI7000, LiCOR), calibrated for accuracy with certified reference material[90] at a typical precision of better than 2 μmol kg$^{-1}$ [91]. All samples were processed immediately, stored on ice while the survey was underway, and frozen (−20 °C) as soon as possible (typically within five hours of collection) except for DIC, which was stored at room temperature. On the main research vessel and the smaller boat, water current velocity was measured using a differential GPS-assisted Lagrangian method with a neutrally-buoyant drifter (adapted from Wetzel and Likens[92]). Current velocity measurements likely indicated flow rates of the ebbing tide, as the surveys were carried out after the turn of the high tide. Water depth on the main research boat was measured using a hull-mounted acoustic transducer (Airmar), while water depth was measured using a lead and line on the smaller boat or taken from Roper et al.[47]. Barometric pressure (±0.5hPa @20 °C), air temperature (±1.1 °C @20 °C), and true wind speed (±5% @10 m s$^{-1}$) were measured 3 m above the water surface using a vessel-mounted weather station (200WX, Airmar). In the ICOLL lagoons, daily averaged meteorological data were obtained from the closest Bureau of Meteorology (BOM) weather station (Climate Data Online)[93].

## Water-air CO$_2$ flux calculations
Water-air CO$_2$ flux ($F$CO$_2$; mmol CO$_2$-C m$^{-2}$ d$^{-1}$) was calculated at 1-minute intervals using Eq. (1):

$$F = k_{600}K_0(C_{water} - C_{air}) \tag{1}$$

where $k_{600}$ is the gas transfer velocity (m d$^{-1}$), $K_0$ is the solubility coefficient of CO$_2$ (mol l$^{-1}$ atm$^{-1}$), and $C_{water}$ and $C_{air}$ are the partial pressure of CO$_2$ (μatm) in water and air, respectively[94]. The formula from Weiss[95] was used to obtain CO$_2$ solubility coefficients based on salinity and temperature Eq. (2):

$$\ln K_0 = A_1 + A_2\left(\frac{100}{T}\right) + A_3 \ln\left(\frac{T}{100}\right) + S\left[B_1 + B_2\left(\frac{T}{100}\right) + B_3\left(\frac{T}{100}\right)^2\right] \tag{2}$$

where $K_0$ is expressed in moles L$^{-1}$ atm$^{-1}$, $A_1$ (-58.0931), $A_2$ (90.5069), $A_3$ (22.2940), $B_1$ (0.027766), $B_2$ (-0.025888), and $B_3$ (0.0050578) are constants, $T$ is absolute temperature, and $S‰$ is the salinity. CO$_2$ atmospheric concentration was assumed to be 407 μatm[96], which was the mean concentration in 2018. Although $k_{600}$ is a significant variable required for calculating water-air fluxes, measuring $k_{600}$ in-situ was not feasible due to the large spatial coverage of this study. As such, five empirical $k_{600}$-models for a range of coastal-marine ecosystems were used from the literature to estimate mean $k_{600}$ (equations (6) to (10) listed in Table 4), i.e., mangrove-dominated[28,97] and tidal[27] (using wind speed, water depth, and current velocity), lagoonal[53] (using wind speed and water depth), and marine-dominated[94] (using wind speed only) coastal ecosystems. Windspeed is corrected for a height of 10 m ($U_{10}$) by rearranging the formula

**Table 4 | Gas transfer velocity normalised to Schmidt number of 600 ($k_{600}$) parameterisations using various methods in published literature**

| Literature | $k_{600}$ parameterisations | Method | Study area | Eqn. |
|---|---|---|---|---|
| Rosentreter et al.[97] | $k_{600} = -0.08 + 0.26\,v + 0.83U_{10} + 0.59\,h$ | Flux chamber | Three mangrove estuaries | 6 |
| Borges et al.[27] | $k_{600} = 1 + 1.719v^{0.5}\,h^{-0.5} + 2.58U_{10}$ | Flux chamber | Macrotidal estuary (Scheldt) | 7 |
| Jiang et al.[53] | $k_{600} = 0.314\,U_{10}^2 - 0.436U_{10} + 3.99$ | Predictive modelling | Global | 8 |
| Ho et al.[28]. | $k_{600} = 0.77\,v^{0.5}\,h^{-0.5} + 0.266U_{10}$ | $^3$He/SF$_6$ | Mangrove estuaries | 9 |
| Wanninkhof[94] | $k = 0.251U_{10}^2\,(Sc/660)^{-0.5}$ | Global ocean inverse model | Global | 10 |

$v$ is the current velocity in cm s$^{-1}$, $h$ is the water depth (m), and $U_{10}$ is the wind speed (m s$^{-1}$) at 10 m height calculated according to Amorocho and DeVries[98].

from Amorocho and DeVries[98]:

$$U_z = U_{10}\left[1 - \frac{(C_{10})^{\frac{1}{2}}}{k}\ln\left(\frac{10}{z}\right)\right] \qquad (3)$$

where $U_z$ is the measured windspeed at $z$ height (3 m) in m s$^{-1}$, $C_{10}$ is the surface drag coefficient for wind at 10 m ($1.3 \times 10^{-3}$[97]), and $\kappa$ is the Von Karman constant (0.41). In the first four parameterisations (Eqs. (6) to (9) in Table 4), $k_{600}$ is the gas transfer velocity (cm h$^{-1}$) normalised to a Schmidt number of 600. The parameterisation in equation 10 (Table 4) by Wanninkhof[94] calculated $k$ at the Schmidt number (Sc) of the measured temperature and salinity, converted to $k_{600}$ using Eq. (4):

$$k_{600} = k\left(\frac{600}{Sc}\right)^{-0.5} \qquad (4)$$

A Schmidt exponent of -0.5 was used to account for higher water turbulences associated with tidal currents in estuaries[99]. To calculate water-air $CO_2$ fluxes, $k$ was derived from $k_{600}$, which was calculated using the other four parameterisations (Eqs. (6) to (9) in Table 4) by rearranging Eq. (4). Sc at the measured temperature and salinity was calculated using the formula in Wanninkhof[94] (Eq. (5)):

$$Sc = A + Bt + Ct^2 + dt^3 + Et^4 \qquad (5)$$

where $A$, $B$, $C$, $D$, and $E$ are constants for $CO_2$ in fresh water (1923.6, −125.06, 4.3773, −0.085681, and 0.00070284, respectively) and seawater (2116.8, −136.25, 4.7353, −0.092307, 0.0007555, respectively), and $t$ is temperature in °C. A salinity factor was calculated from the difference between freshwater and sea water Sc and applied to calculate Sc at the measured salinity.

To ensure consistency between water-air $CO_2$ fluxes measured for estuaries in this study and those previously reported for eight southeast Queensland estuaries[12], water-air $CO_2$ fluxes were recalculated using the five parameterisations. Water-air $CO_2$ fluxes from the previously reported three north Queensland estuaries[30] were not recalculated because water depth and current velocity data were unavailable. However, given that these three estuaries were each categorised in a different disturbance group and/or estuary type (moderate and high disturbance tidal system, and a high disturbance small delta, Table 3), this should not introduce any systematic bias.

**Data processing and statistical analysis**
Per-minute $pCO_2$ and water-air $CO_2$ flux were averaged to 5-minute datapoints to reduce the number of data points whilst maintaining the high resolution and main features of the dataset. Kolmogorov-Smirnov and Levene's tests for normality and homoscedasticity, respectively, returned significant results ($p < 0.05$), ruling out parametric methods for statistical analysis. Consequently, significant differences ($\alpha = 0.05$) between and within estuary types (3 factors) and disturbance groups

(4 factors) were tested using the Permutational Multivariate Analysis of Variance (PERMANOVA) procedure with Euclidean distance as the dissimilarly matrix in Primer v7 and PERMANOVA+ add-on (PRIMER-e). The dataset for PERMANOVA analysis was normalised ($z$-score) but not power-transformed to retain the heterogeneity between the mean and variances, retaining the spatial scale along the estuarine transect. Not power-transforming the data reduces the possibility of an inflated type I error[100]. Salinity was included as a covariate in the $pCO_2$, water-air $CO_2$ flux, DOC and DIC analyses to remove differences influenced by salinity. The focus of PERMANOVA is on the differences between the data points rather than descriptive statistics (mean, median, etc.). PERMANOVA can therefore, identify significant differences in datasets even where there are similarities in the descriptive statistics. 9999 permutations were performed using residuals under a reduced model using type I sum of squares. Significant results were further analysed with pairwise PERMANOVA. $pCO_2$ and water-air $CO_2$ fluxes were analysed for correlations with salinity using Pearson's correlation and combined with physicochemical data, DOC, DIC, and percent seagrass cover to analyse for partial correlations while controlling for the effect of salinity ($\alpha = 0.05$) in SPSS v25 (IBM). Data used for correlation analysis were power-transformed where necessary and normalised ($z$-score). Partial correlation analysis was chosen over multivariate methods such as Principal Component Analysis (PCA) because targeted testing for correlations between variables and $CO_2$ was more useful than an exploratory approach.

**$CO_2$ emission upscaling to the Australian continent**
Published summer and winter water-air $CO_2$ flux rates were available for 13 Australian estuaries, including each of the three geomorphic estuary types (Supplementary Table 1). Of these, 10 of the estuaries were included in this study[12,30], along with an additional three from a published study[43]. The summer and winter water-air $CO_2$ fluxes were used to calculate a summer:winter water-air $CO_2$ flux ratio (mean and range) for each of the three estuary types (Supplementary Table 2). Ratios for each estuary type were then applied to the measured summer water-air $CO_2$ fluxes for the 47 estuaries to estimate the mean and range of winter water-air $CO_2$ fluxes (Table 2). The summer and winter mean water-air $CO_2$ flux rates from each estuary were averaged together to derive the annual water-air $CO_2$ flux rates and emissions for the 47 estuaries. To gauge the sensitivity of annual Australian emissions to winter flux rates, the summer mean flux rates were also adjusted up and down by the minimum and maximum of winter:summer ratios (Supplementary Table 2). Upscaled fluxes determined based on these minimum and maximum ratios allow an upper and lower limit to be placed on these estimates.

Annual $CO_2$ emissions from the 47 estuaries were upscaled to all Australian estuaries ($n = 971$) by multiplying the estuary type-specific and disturbance-specific water-air $CO_2$ fluxes (mmol $CO_2$-C m$^{-2}$ d$^{-1}$) by the total estuarine surface area of the relevant systems[40] (Table 3). Small deltas with low to moderate disturbance were not available for this study. However, even though measured low and moderately disturbed small delta water-air $CO_2$ fluxes were likely different from the

mean high and very high disturbed small delta water-air $CO_2$ flux, their impact on total Australian estuary emissions is low. This is because small deltas only make up 1.5% of Australia's estuarine surface area (Table 3).

**Reporting summary**

Further information on research design is available in the Nature Portfolio Reporting Summary linked to this article.

## Data availability

The environmental survey data generated/used in this study is freely available and has been deposited in the FigShare database under accession code https://doi.org/10.6084/m9.figshare.25242676. Figure source data are provided with this paper as part of the Supplementary Information. Source data are provided with this paper.

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

## Acknowledgements

This research was funded by Australian Research Council grants DP160100248 (BDE) and LP150100519 (BDE). We would like to thank Western Australia's (WA) Department of Water and Environmental Protection (DWER) for the logistical support and scientific knowledge provided for the WA estuary surveys.

## Author contributions

All authors have agreed to be listed and have approved the submitted version of the manuscript. JY conceived the project, collected data, ran data analysis and interpretation, and led the writing of the manuscript. JR and JO collected data, contributed to interpretation, and helped write the manuscript. BE conceived the project, collected data, contributed to interpretation, and helped write the manuscript. KS contributed to data analysis, and helped write the manuscript.

## Competing interests
The authors declare no competing interests.
