## [Peer Review File · Nature Communications]

REVIEWER COMMENTS

Reviewer #1 (Remarks to the Author):

This manuscript focuses on measuring carbon dioxide emissions from 47 Australian estuaries based on dissolved CO₂ surveys and scaling up these measurements based on another study that measured CO₂ emissions over two seasons, to all 974 Australia estuaries. This report is well written, helps quantify an important carbon cycle component in Australia, and is conducted across an impactful spatial scale.

This study reports on noteworthy results. The authors found that disturbed estuaries - or those with more developed watersheds - were associated with greater carbon dioxide emissions. In addition, estuaries that were drowned river valleys or tidal embayments emitted more CO₂ than those than lagoons and small river deltas.

There several things that need to be improved prior to publication. First, the main finding of the report is that CO₂ emissions are a function of estuarine geomorphology. However, one of the main "types" of estuarine categories reported on is a "tidal system." This is defined in the methods section as a drowned river valley or tidal embayment; however these do not seem like synonymous types of estuaries. I suggest that the authors define the types of estuaries better, by referring to a source that delineates a "tidal system" category and do it earlier in the paper. Alternatively, separate bays and drowned river valleys could be put into different categories. Or, perhaps the estuarine classification should be separated by tidal range. In summary, "tidal system" does not seem to be a defensible and consistent category, and more thought should go into reporting the relationship between CO₂ emissions and the type of estuary.

A similar issue has to do with the disturbance categories which need to be described in more detail earlier in the paper. It is helpful that a source is reported, but more detail is needed about how the classification was performed, perhaps in the supplemental material.

In addition, one of the main focal areas of the paper is on the relation between CO₂ and emission and estuarine geomorphology. However, there is great covariability in the "tidal system" category with latitude, and presumably, temperature, as most of the the "tidal systems" are in the north. Can the higher CO₂ emissions in the tidal system category be linked to temperature, to tidal range, or to the structure of the estuary? This question should be addressed through data analysis, where the authors consider for instance an interaction term between latitude (as a surrogate for climate) and estuarine type.

In terms of the level of detail provided regarding methods, there are several areas where more details are needed, perhaps in supplementary sections. Additional detail about how the error analysis was conducted and how the upscaling was done need to be incorporated. In addition, the paper mentions that in some cases the CO₂ surveys were done by kayak, but no detail is given. Was the instrumentation taken by kayak through the estuaries, which doesn't seem too probable, or were water samples collected by kayak and analyzed shipboard?

Other methodological questions: did CO₂ levels vary over the course of the day, and would this factor affect upscaled estimates? Would CO₂ emissions over the course of a year vary in a way not captured by winter and summer estimates?

Also useful context could be what seasonal cycles look like in CO₂ in air and ocean waters in the southern hemisphere. Coastal estuaries emit land derived CO₂, but oceanic processes can also be important.

Overall, this paper is well written and high impact, and urge that it be published, but considerable revisions are needed prior to publication.

Reviewer #3 (Remarks to the Author):

Please see attached comments

Reviewer comments for ‘High carbon dioxide emissions from Australian estuaries driven by geomorphology’

This paper provides a comprehensive overview of carbon dioxide emissions from three distinct estuarine typologies found in Australia and then employs upscaling to inform the national importance of this emissions term. In doing so it redresses the northern hemisphere bias in observations of carbon dioxide emissions from estuaries, thereby making an important contribution to the literature. The manuscript is well written and the figures are clear (with the exception of Figure 1 which is poorly resolved). However I do have concerns surrounding the novelty of the research and the degree to which the conclusions are supported by the presented data.

Geomorphic type

There are a number of statements throughout the manuscript that highlight that geomorphic type controls surface area which in turn dictates emissions source size. Surface area controlling emissions is not a novel finding, and although necessary to produce national budgets, and enable global comparisons, references to total surface area controlling emissions source size should be minimised in favour of understanding area-weighted emissions, particularly as the paper does not try to refine surface area estimates for Australian estuaries.

After reading the paper I'm not sure I agree with the title. The emissions are higher than average because Australia has more mangrove dominated tropical and subtropical estuaries which are absent in well studied temperate northern hemisphere systems. Does this not imply that climate setting is the dominant control rather than geomorphic type?

Lateral inputs

Lateral inputs are poorly defined and there is little data pertaining to their importance. This should be addressed particularly as one of the main conclusions is that Australian deltas and tidal systems have higher than average global emissions due to lateral inputs. Does this term include inputs of CO₂ supersaturated water and/or organic matter? Where is the evidence that lateral inputs are linked to emissions? Tidal range is inferred to control lateral inputs but I can't see any specific evidence linking emissions and lateral inputs. I think more data is needed for this conclusion to remain.

Disturbance effects in lagoons

The paper states that both pCO₂ and water-air CO₂ flux in lagoons increased significantly with higher disturbance but only from low to high disturbance, and was significantly lower in very high disturbance lagoons compared to high disturbance lagoons. There is a reference in the discussion suggesting that oxygen saturations altered the very high disturbance system to a phytoplankton dominated state (thereby enabling enhanced carbon dioxide drawdown?). However, this finding is not explored sufficiently and is only implied as a causal mechanism. It is important for the reader to understand this non-linear disturbance response given the presented interplay between geomorphic type and disturbance (with lagoons different to deltas and tidal systems in their response to disturbance), and the observation that there are more disturbed lagoon systems in the northern hemisphere, thereby providing a basis for comparison with existing studies. I understand that the definitions used to categorise disturbance are provided in the supplementary information, but these are entirely qualitative and it would be more insightful to present explanatory data (e.g. physico-chemical parameters, nutrients, heavy metals) to help interpret the non-linear disturbance effect observed for lagoons.

Replies to Reviewers

We thank the reviewers for their helpful and insightful comments that helped improve the manuscript.

REVIEWER COMMENTS:

Reviewer #1 (Remarks to the Author):

This manuscript focuses on measuring carbon dioxide emissions from 47 Australian estuaries based on dissolved CO₂ surveys and scaling up these measurements based on another study that measured CO₂ emissions over two seasons, to all 974 Australia estuaries. This report is well written, helps quantify an important carbon cycle component in Australia, and is conducted across an impactful spatial scale.

This study reports on noteworthy results. The authors found that disturbed estuaries - or those with more developed watersheds - were associated with greater carbon dioxide emissions. In addition, estuaries that were drowned river valleys or tidal embayments emitted more CO₂ than those than lagoons and small river deltas.

COMMENT 1:

The main finding of the report is that CO₂ emissions are a function of estuarine geomorphology. However, one of the main "types" of estuarine categories reported on is a "tidal system." This is defined in the methods section as a drowned river valley or tidal embayment; however these do not seem like synonymous types of estuaries. I suggest that the authors define the types of estuaries better, by referring to a source that delineates a "tidal system" category and do it earlier in the paper. Alternatively, separate bays and drowned river valleys could be put into different categories. Or, perhaps the estuarine classification should be separated by tidal range. In summary, "tidal system" does not seem to be a defensible and consistent category, and more thought should go into reporting the relationship between CO₂ emissions and the type of estuary.

REPLY:

We have used the 'tidal system' estuary type that is commonly used in the literature for regional and global classifications (e.g., Dürr et al. 2011, Laruelle et al, 2013, Rosentreter et al. 2023). Definitions of the estuary types is provided in Supplementary Table 2. However, this Table was not referenced in the methods section. This has now been added (L431 and 435). In previous classifications, drowned river valleys and tidal embayments are both classified as tidal systems (Dürr et al. 2011). We have therefore used this classification so that Australian estuaries are comparable with their respective estuary types globally.

COMMENT 2:

A similar issue (continuing from comment 1) has to do with the disturbance categories which need to be described in more detail earlier in the paper. It is helpful that a source is reported, but more detail is needed about how the classification was performed, perhaps in the supplemental material.

REPLY:

We thank the reviewer for his comment. The classification criteria used to define the disturbance categories was provided in Supplementary Table 1. We have added some

additional details in the methods section to further clarify how these disturbance categories were applied (L420 to 427).

COMMENT 3:

In addition, one of the main focal areas of the paper is on the relation between CO₂ and emission and estuarine geomorphology. However, there is great covariability in the "tidal system" category with latitude, and presumably, temperature, as most of the the "tidal systems" are in the north. Can the higher CO₂ emissions in the tidal system category be linked to temperature, to tidal range, or to the structure of the estuary? This question should be addressed through data analysis, where the authors consider for instance an interaction term between latitude (as a surrogate for climate) and estuarine type.

REPLY:

Comparison of temperature to *p*CO₂ and water-air CO₂ flux, while factoring for the salinity effect, is presented in the manuscript's Supplementary information section, and showed no correlation. We have further added a section showing significant correlations for tidal range and CO₂ across the three estuary types (L920 to 921). We have also added more details to the discussion of temperature and tidal range effects on CO₂ (L220 to 235). Our results show that tidal range (and the associated lateral exchange) and not temperature, was likely the significant driver for the differences in CO₂ results between estuary types. Although we discuss the importance of climate on CO₂ emissions (and biogeochemistry) of Australian estuaries, it was not feasible to include climate as a variable in our study due to the large number of estuaries required (3 estuary types x 4 disturbance groups x at least 3 climate types x replication of each type).

COMMENT 4:

In terms of the level of detail provided regarding methods, there are several areas where more details are needed, perhaps in supplementary sections. Additional detail about how the error analysis was conducted and how the upscaling was done need to be incorporated. In addition, the paper mentions that in some cases the CO₂ surveys were done by kayak, but no detail is given. Was the instrumentation taken by kayak through the estuaries, which doesn't seem too probable, or were water samples collected by kayak and analysed shipboard?

REPLY:

Methodological details of error analysis and annual emission upscaling, are in the methods section 'CO₂ emission upscaling to the Australian continent' (L566 to 578). We have added additional details of how these were done and also restructured some sentences to improve clarity. In both the main research vessel and the smaller boat (originally called kayak), discrete samples were collected during the survey and stored accordingly onboard. Water current measurements were done similarly on both vessels, but water depth measurements were done using a lead and line on the smaller boat compared to an acoustic transducer on the main research boat (L505 to 520). In the smaller boat used in ICOLL lagoons, we do not have the Picarro CRDS analyser running but still have the LiCOR CO₂ analyser measuring continuously (L482 to 484). These details have now been added and clarified in the manuscript.

COMMENT 5:

Other methodological questions: did CO₂ levels vary over the course of the day, and would this factor affect upscaled estimates? Would CO₂ emissions over the course of a year vary in a way not captured by winter and summer estimates?

REPLY:

Our surveys were only undertaken during the day, and not overnight, and therefore do not capture diurnal changes. Although there can be some variations over the diurnal cycle, it is more than likely that diurnal variations were the result of other influences such as tidal variations, riverine input, and lateral exchange with intertidal zones (Yang et al. 2017, Polsenaere et al. 2023, and de la Paz et al. 2007). We account for seasonal variation between the summer and winter extremes, but not episodic events that can significantly increase the input of CO₂ saturated waters into estuaries (Ruiz-Halpern et al. 2015). However, the focus of our paper was to provide the first estimate of CO₂ emissions from Australia accounting for estuarine geomorphology and disturbance. As such, we sampled a large number of estuaries under similar conditions, and it was not possible to consider the above factors. We have added some discussion of these points (L258 to 268).

COMMENT 6:

Also useful context could be what seasonal cycles look like in CO₂ in air and ocean waters in the southern hemisphere. Coastal estuaries emit land derived CO₂, but oceanic processes can also be important.

REPLY:

The CO₂ seasonal cycle in the atmospheric (Friedlingstein et al. 2022) and coastal waters (Schulz et al. 2019, Roobaert et al. 2019) in the southern hemisphere is much smaller than CO₂ changes along the estuaries, suggesting little influence on estuarine emissions.

COMMENT 7:

Overall, this paper is well written and high impact, and urge that it be published, but considerable revisions are needed prior to publication.

REPLY:

We thank the reviewer for their positive and insightful comments.

Reviewer #3 (Remarks to the Author):

COMMENT 1:

This paper provides a comprehensive overview of carbon dioxide emissions from three distinct estuarine typologies found in Australia and then employs upscaling to inform the national importance of this emissions term. In doing so it redresses the northern hemisphere bias in observations of carbon dioxide emissions from estuaries, thereby making an important contribution to the literature. The manuscript is well written and the figures are clear (with the exception of Figure 1 which is poorly resolved).

REPLY:

We thank the reviewer for their positive comments. We have improved the resolution of Figure 1 (L875).

COMMENT 2:

Re: Geomorphic type

There are a number of statements throughout the manuscript that highlight that geomorphic type controls surface area which in turn dictates emissions source size. Surface area controlling emissions is not a novel finding, and although necessary to produce national budgets, and enable global comparisons, references to total surface area controlling emissions source size should be minimised in favour of understanding area-weighted emissions, particularly as the paper does not try to refine surface area estimates for Australian estuaries.

REPLY:

We agree that surface area is not the direct driver for CO₂ emissions, although it is the indirect result of geomorphological effects on CO₂ emissions. We have now removed the statements of estuary type surface area controlling CO₂ emissions in the manuscript.

COMMENT 3:

Re: Geomorphic type

After reading the paper I'm not sure I agree with the title. The emissions are higher than average because Australia has more mangrove dominated tropical and subtropical estuaries which are absent in well studied temperate northern hemisphere systems. Does this not imply that climate setting is the dominant control rather than geomorphic type?

REPLY:

We agree that climate is an important control of CO₂ emissions, similarly to estuary geomorphic type. We have added climate as a driving factor in the title.

COMMENT 4:

Re: Lateral inputs

Lateral inputs are poorly defined and there is little data pertaining to their importance. This should be addressed particularly as one of the main conclusions is that Australian deltas and tidal systems have higher than average global emissions due to lateral inputs. Does this term include inputs of CO₂ supersaturated water and/or organic matter? Where is the evidence that lateral inputs are linked to emissions? Tidal range is inferred to control lateral inputs but I can't see any specific evidence linking emissions and lateral inputs. I think more data is needed for this conclusion to remain.

REPLY:

In a regional scale study, we are only able to provide evidence linking emissions to lateral inputs through tidal range as a proxy for the magnitude of lateral export. As such, we have reduced the emphasis on lateral exports as a direct driver for emissions (L221 to 233).

COMMENT 5:

Re: Disturbance effects in lagoons

The paper states that both pCO₂ and water-air CO₂ flux in lagoons increased significantly with higher disturbance but only from low to high disturbance, and was significantly lower in very high disturbance lagoons compared to high disturbance lagoons. There is a reference in the discussion suggesting that oxygen saturations altered the very high disturbance system to a phytoplankton dominated state (thereby enabling enhanced carbon dioxide drawdown?). However, this finding is not explored sufficiently and is only implied as a causal mechanism. It is important for the reader to understand this non-linear disturbance response given the presented interplay between geomorphic type and disturbance (with lagoons different to deltas and tidal systems in their response to disturbance), and the observation that there are more disturbed lagoon systems in the northern hemisphere, thereby providing a basis for comparison with existing studies. I understand that the definitions used to categorise disturbance are provided in the supplementary information, but these are entirely qualitative and it would be more insightful to present explanatory data (e.g. physico-chemical parameters, nutrients, heavy metals) to help interpret the non-linear disturbance effect observed for lagoons.

REPLY:

We agree that this non-linear response is interesting. However, to explore this in more detail would be speculative with the regional scale data we have. We have therefore made no change in response to this comment.

ADDITIONAL CHANGES:

In addition to responding to the comments by the reviewers, we have now further added previously missing surface area measurements for 108 estuaries (of 971 estuaries), resulting in updated total Australian estuarine surface area figures and CO₂ emission estimates. These have been incorporated into the manuscript. Changes for updated surface area and CO₂ emission are found in L12-13, L72, 74, 85, 154-156, 164-165, 167-190, 313-327, 359-379, 436-437, Tables 1 and 3, Figure 5, and Supplementary tables 14.

Standard error (SE) was also used rather than standard deviation (SD) when calculating for mean categorical CO₂ emissions (estuary type and disturbance groups). This is because SE gives a gauge on how accurate our estimates are whereas SD which measures the dispersion of measurements in the estuaries (which naturally do). Changes were made in L12, 114-117, 129-130, 154, 313-322, 359-378, Tables 2 and 3, and Supplementary Tables 9, 11, and 13.

Units for water-air flux were also changed to be in-line with other studies in the literature in L272-278, Table 3, Figure 5, and Supplementary Table 4.

REFERENCES:

- Yang, W. Bin, Yuan, C.S., Tong, C., Yang, P., Yang, L., Huang, B.Q., 2017. Diurnal variation of CO₂, CH₄, and N₂O emission fluxes continuously monitored in-situ in three environmental habitats in a subtropical estuarine wetland. *Mar. Pollut. Bull.* 119, 289–298.
- Polsenaere, P., Delille, B., Poirier, D., Charbonnier, C., Deborde, J., Mouret, A., Abril, G., 2023. Seasonal, Diurnal, and Tidal Variations of Dissolved Inorganic Carbon and pCO₂ in Surface Waters of a Temperate Coastal Lagoon (Arcachon, SW France). *Estuaries and Coasts* 46, 128–148.
- de la Paz, M., Gómez-Parra, A., Forja, J., 2007. Inorganic carbon dynamic and air-water CO₂ exchange in the Guadalquivir Estuary (SW Iberian Peninsula). *J. Mar. Syst.* 68, 265–277.
- de la Paz, M., Gómez-Parra, A., Forja, J., 2007. Inorganic carbon dynamic and air-water CO₂ exchange in the Guadalquivir Estuary (SW Iberian Peninsula). *J. Mar. Syst.* 68, 265–277.
- Friedlingstein, P., O’Sullivan, M., Jones, M.W., Andrew, R.M., Gregor, L., Hauck, J., Le Quéré, C., Luijkx, I.T., Olsen, A., Peters, G.P., Peters, W., Pongratz, J., Schwingshackl, C., Sitch, S., Canadell, J.G., Ciais, P., Jackson, R.B., Alin, S.R., Alkama, R., Arneeth, A., Arora, V.K., Bates, N.R., Becker, M., Bellouin, N., Bittig, H.C., Bopp, L., Chevallier, F., Chini, L.P., Cronin, M., Evans, W., Falk, S., Feely, R.A., Gasser, T., Gehlen, M., Gkritzalis, T., Gloege, L., Grassi, G., Gruber, N., Gürses, Ö., Harris, I., Hefner, M., Houghton, R.A., Hurtt, G.C., Iida, Y., Ilyina, T., Jain, A.K., Jersild, A., Kadono, K., Kato, E., Kennedy, D., Klein Goldewijk, K., Knauer, J., Korsbakken, J.I., Landschützer, P., Lefèvre, N., Lindsay, K., Liu, J., Liu, Z., Marland, G., Mayot, N., McGrath, M.J., Metzl, N., Monacchi, N.M., Munro, D.R., Nakaoka, S.-I., Niwa, Y., O’Brien, K., Ono, T., Palmer, P.I., Pan, N., Pierrot, D., Pockock, K., Poulter, B., Resplandy, L., Robertson, E., Rödenbeck, C., Rodriguez, C., Rosan, T.M., Schwinger, J., Séférian, R., Shutler, J.D., Skjelvan, I., Steinhoff, T., Sun, Q., Sutton, A.J., Sweeney, C., Takao, S., Tanhua, T., Tans, P.P., Tian, X., Tian, H., Tilbrook, B., Tsujino, H., Tubiello, F., van der Werf, G.R., Walker, A.P., Wanninkhof, R., Whitehead, C., Willstrand Wranne, A., Wright, R., Yuan, W., Yue, C., Yue, X., Zaehle, S., Zeng, J., Zheng, B., 2022. Global Carbon Budget 2022. *Earth Syst. Sci. Data* 14, 4811–4900.
- Schulz, K.G., Hartley, S., Eyre, B., 2019. Upwelling Amplifies Ocean Acidification on the East Australian Shelf: Implications for Marine Ecosystems. *Front. Mar. Sci.* 6, 1–8.
- Roobaert, A., Laruelle, G.G., Landschützer, P., Gruber, N., Chou, L., Regnier, P., 2019. The Spatiotemporal Dynamics of the Sources and Sinks of CO₂ in the Global Coastal Ocean. *Global Biogeochem. Cycles* 33, 1693–1714.
- Dürr, H.H., Laruelle, G.G., van Kempen, C.M., Slomp, C.P., Meybeck, M., Middelkoop, H., 2011. Worldwide typology of nearshore coastal systems: Defining the estuarine filter of river inputs to the oceans. *Estuaries and Coasts* 34, 441–458.

Laruelle, G.G., Dürr, H.H., Lauerwald, R., Hartmann, J., Slomp, C.P., Goossens, N., Regnier, P.A.G., 2013. Global multi-scale segmentation of continental and coastal waters from the watersheds to the continental margins. *Hydrol. Earth Syst. Sci.* 17, 2029–2051.

Rosentreter, J.A., Laruelle, G.G., Bange, H.W., Bianchi, T.S., Busecke, J.J.M., Cai, W., Eyre, B.D., Forbrich, I., Kwon, E.Y., Maavara, T., Moosdorf, N., Najjar, R.G., Sarma, V.V.S.S., Van Dam, B., Regnier, P., 2023. Coastal vegetation and estuaries collectively are a greenhouse gas sink. *Nat. Clim. Chang.*

REVIEWERS' COMMENTS

Reviewer #3 (Remarks to the Author):

I have no further comments.

Reviewer #4 (Remarks to the Author):

General comments:

This study made detailed measurements of CO₂ emissions across a subset of Australian estuaries (roughly 5% of the estuaries around the continent) and scaled these emissions to the entire continent based on geomorphological classifications. The team found Australian estuarine emissions to be higher than the global average, in part due to the climate region as well as a lack of disturbance and the geomorphological makeup of the estuaries.

The study is both important and well-written. I did not review the first version of this manuscript but assessed the authors' response to the first round of reviews. The initial reviewers provided numerous important points to consider, which the authors have done a good job addressing in my opinion.

Perhaps the biggest consideration is how to robustly scale emissions from ~5% of the observed estuaries to the other 95% of the estuaries without any observations. Using geomorphological classification seems like a reasonable approach to me, particularly for a "first stab" at scaling emissions across the continent. This is certainly an improvement over how Australian estuaries are represented in current global CO₂ budgets and will provide the community a good starting point to improve on. Likewise, this approach can be used as an example to follow for other regions that are underrepresented in terms of observational data. Other approaches one could use are tools such as machine learning models that would consider any available observational data as explanatory variables (e.g., meteorology, terrestrial land use, river discharge, water physiochemistry, etc.). However, these approaches often provide overconfidence in model performance based on spurious correlations with factors that have nothing to do with CO₂ emissions. For this reason, I appreciate the fairly simple approach taken here.

Overall, I am supportive of publication of this manuscript. And considering the author's thoughtful response to previous reviews, I have not made highly detailed comments, but have provided a few specific suggestions to consider below.

Specific Comments:

Lines 52-69: This paragraph discusses residence time of different estuarine types and attributes

estuarine CO₂ emissions primarily to decomposition of DOC within the estuary. Consider also adding a sentence noting that in large river dominated estuaries, a large fraction of CO₂ emissions can also be associated with CO₂ exported from the river system itself. For example, see:

Valerio, A.M., Kampel, M., Ward, N.D., Sawakuchi, H.O., Cunha, A.C., Krusche, A.V., Richey, J.E. (2021) CO₂ partial pressure and fluxes in the Amazon River Plume using in situ and remote sensing data. *Continental Shelf Research*. 215, 104348. <https://doi.org/10.1016/j.csr.2021.104348>

In the case of the Amazon River estuary noted above, CO₂ export from the river and subsequent emission in the nearshore estuary can nearly balance CO₂ uptake in the offshore estuary. This concept was briefly discussed in the discussion around line 219, but is worth mentioning up front in the introduction as well since this paragraph summarized mechanisms of estuarine CO₂ dynamics.

Line 229: The higher CO₂ is indirect evidence of lateral inputs. Perhaps you could change this to “but, we did not directly measure lateral inputs.”

Line 331-371: This is a very long paragraph with at least three different topics discussed. Consider adding a break at line 337, which switches to a new topic (deltas and tidal systems). Then again at line 350, which compares geomorphology to climate effects.

Line 608: Please provide a doi link to the figshare dataset to increase findability, accessibility, interoperability, and reusability.

REVIEWERS' COMMENTS

Reviewer #3 (Remarks to the Author):

I have no further comments.

Reviewer #4 (Remarks to the Author):

General comments:

This study made detailed measurements of CO₂ emissions across a subset of Australian estuaries (roughly 5% of the estuaries around the continent) and scaled these emissions to the entire continent based on geomorphological classifications. The team found Australian estuarine emissions to be higher than the global average, in part due to the climate region as well as a lack of disturbance and the geomorphological makeup of the estuaries.

The study is both important and well-written. I did not review the first version of this manuscript but assessed the authors' response to the first round of reviews. The initial reviewers provided numerous important points to consider, which the authors have done a good job addressing in my opinion.

Perhaps the biggest consideration is how to robustly scale emissions from ~5% of the observed estuaries to the other 95% of the estuaries without any observations. Using geomorphological classification seems like a reasonable approach to me, particularly for a "first stab" at scaling emissions across the continent. This is certainly an improvement over how Australian estuaries are represented in current global CO₂ budgets and will provide the community a good starting point to improve on. Likewise, this approach can be used as an example to follow for other regions that are underrepresented in terms of observational data. Other approaches one could use are tools such as machine learning models that would consider any available observational data as explanatory variables (e.g., meteorology, terrestrial land use, river discharge, water physiochemistry, etc.). However, these approaches often provide overconfidence in model performance based on spurious correlations with factors that have nothing to do with CO₂ emissions. For this reason, I appreciate the fairly simple approach taken here.

Overall, I am supportive of publication of this manuscript. And considering the author's thoughtful response to previous reviews, I have not made highly detailed comments, but have provided a few specific suggestions to consider below.

Specific Comments:

COMMENT:

Lines 52-69: This paragraph discusses residence time of different estuarine types and attributes

estuarine CO₂ emissions primarily to decomposition of DOC within the estuary. Consider also adding a sentence noting that in large river dominated estuaries, a large fraction of CO₂ emissions can also be associated with CO₂ exported from the river system itself. For example, see:

Valerio, A.M., Kampel, M., Ward, N.D., Sawakuchi, H.O., Cunha, A.C., Krusche, A.V., Richey, J.E. (2021) CO₂ partial pressure and fluxes in the Amazon River Plume using in situ and remote sensing data. *Continental Shelf Research*. 215, 104348. <https://doi.org/10.1016/j.csr.2021.104348>

In the case of the Amazon River estuary noted above, CO₂ export from the river and subsequent emission in the nearshore estuary can nearly balance CO₂ uptake in the offshore estuary. This concept was briefly discussed in the discussion around line 219, but is worth mentioning up front in the introduction as well since this paragraph summarized mechanisms of estuarine CO₂ dynamics.

REPLY:

The reviewer makes a valid point regarding the role of estuarine plumes in large river systems. This point was left out initially because large river systems (as classified by Dürr et al., 2011) do not occur in Australia. However, as pointed out by the reviewer, readers might question whether this aspect was considered in the analysis. We have added a statement in L71-74 that addresses this point: “In very large river systems (Dürr et al., 2011) such as the Amazon River in Brazil (Valerio et al., 2021) and the Yangtze River in China (Chen et al., 2012), riverine transport from the estuary to the ocean can result in extensive estuarine plumes that can act as either a source or a sink of CO₂. However, such systems do not exist in Australia.”.

COMMENT:

Line 229: The higher CO₂ is indirect evidence of lateral inputs. Perhaps you could change this to “but, we did not directly measure lateral inputs.”

REPLY:

It appears that the line reference is incorrect as there is no direct mention of lateral inputs at L229. It could be that the comment was for L222. In which case, the suggested edit was added in L225.

COMMENT:

Line 331-371: This is a very long paragraph with at least three different topics discussed. Consider adding a break at line 337, which switches to a new topic (deltas and tidal systems). Then again at line 350, which compares geomorphology to climate effects.

REPLY:

The paragraph is indeed lengthy. We have added breaks at L326 and L339.

COMMENT:

Line 608: Please provide a doi link to the figshare dataset to increase findability, accessibility, interoperability, and reusability.

REPLY:

We have added the doi link ([10.6084/m9.figshare.25242676](https://doi.org/10.6084/m9.figshare.25242676)) to the FigShare repository in L600.